# Molecular mechanism for recognition of the cargo adapter Rab6$^{GTP}$ by the dynein adapter BicD2

Xiaoxin Zhao[1], Sebastian Quintremil[2], Estrella D Rodriguez Castro[1], Heying Cui[1], David Moraga[1], Tingyao Wang[1], Richard B Vallee[2], Sozanne R Solmaz[1]

Rab6 is a key modulator of protein secretion. The dynein adapter Bicaudal D2 (BicD2) recruits the motors cytoplasmic dynein and kinesin-1 to Rab6$^{GTP}$-positive vesicles for transport; however, it is unknown how BicD2 recognizes Rab6. Here, we establish a structural model for recognition of Rab6$^{GTP}$ by BicD2, using structure prediction and mutagenesis. The binding site of BicD2 spans two regions of Rab6 that undergo structural changes upon the transition from the GDP- to GTP-bound state, and several hydrophobic interface residues are rearranged, explaining the increased affinity of the active GTP-bound state. Mutations of Rab6$^{GTP}$ that abolish binding to BicD2 also result in reduced co-migration of Rab6$^{GTP}$/BicD2 in cells, validating our model. These mutations also severely diminished the motility of Rab6-positive vesicles in cells, highlighting the importance of the Rab6$^{GTP}$/BicD2 interaction for overall motility of the multi-motor complex that contains both kinesin-1 and dynein. Our results provide insights into trafficking of secretory and Golgi-derived vesicles and will help devise therapies for diseases caused by BicD2 mutations, which selectively affect the affinity to Rab6 and other cargoes.

## Introduction

Rab6 is a key modulator of protein secretion and exocytosis. It is the most abundant Rab GTPase that is embedded in secretory and Golgi-derived vesicles and serves as an identity marker for these cellular compartments (Goud et al, 1990; Martinez et al, 1994; Grigoriev et al, 2007). Dynein adapters such as Bicaudal D2 (BicD2) have key roles in cellular transport, as they recognize cargoes including Rab6 and link them to dynein motors (Hoogenraad et al, 2001, 2003; Matanis et al, 2002), which are the predominant motors responsible for the transport of cargoes that are directed towards the minus-end of microtubules. BicD2 is important for the transport of secretory and Golgi-derived vesicles and recruits dynein to Rab6$^{GTP}$ (Hoogenraad et al, 2001, 2003).

BicD2 is auto-inhibited in the absence of cargo and unable to recruit dynein because the cargo-binding domain occludes the dynein binding site (Splinter et al, 2012; Liu et al, 2013; Schlager et al, 2014a; McKenney et al, 2014; Terawaki et al, 2015; Urnavicius et al, 2015; McClintock et al, 2018; Sladewski et al, 2018; Cui et al, 2020). Binding of cargo opens up the looped conformation of BicD2 and makes the dynein site accessible. Dynein adapters such as BicD2 also link dynein to its activator dynactin and are thus required for activation of dynein for processive motility. BicD2 thus has a key role in modulating dynein-dependent motility (Splinter et al, 2012; Liu et al, 2013; Schlager et al, 2014a; McKenney et al, 2014; Terawaki et al, 2015; Urnavicius et al, 2015; McClintock et al, 2018; Sladewski et al, 2018; Cui et al, 2020). The motility of dynein in BicD2-dependent transport pathways is further fine-tuned by the opposite polarity motor kinesin-1, which also binds to BicD2 at the coiled-coil domain 2 and impacts overall motility of the motor complex (Grigoriev et al, 2007; Splinter et al, 2010; Serra-Marques et al, 2020).

BicD2 binds to Rab6$^{GTP}$ and recruits dynein to it, but in addition, Rab6$^{GTP}$ also interacts directly with dynein and the p150$_{glued}$ subunit of dynactin (Short et al, 2002; Wanschers et al, 2008). GTP-bound Rab6 is the active form, which is integrated into membranes by prenylation, whereas the inactive Rab6$^{GDP}$ state is released from the membrane by the protein GDP-dissociation inhibitor (GDI) (Goud et al, 1990; Martinez et al, 1994; Grigoriev et al, 2007). Rab6$^{GTP}$ is required for the anterograde transport of vesicles from the medial to trans-Golgi cisterna (Dickson et al, 2020) and also a general modulator of post-Golgi secretion and exocytosis, for which the microtubule motors kinesin-1 (Kif5B) and kinesin-3 (Kif13B) are important (as well as kinesin-3 Kif1C in neurons) (Serra-Marques et al, 2020). For plus end–directed post-Golgi trafficking of the Rab6-containing secretory vesicles, kinesin-1 is the dominant motor and is recruited via BicD2, and kinesin-1 can in addition bind to membranes via the Dopey1-Mon2 complex (Grigoriev et al, 2007; Mahajan et al, 2019; Serra-Marques et al, 2020). Kinesin-1–mediated motility is further fine-tuned by a tug-of-war with the opposite polarity motor BicD2/dynein (Grigoriev et al, 2007; Lee et al, 2015; Serra-Marques et al, 2020). Kinesin-3 (Kif13B) is also associated with

[1]Department of Chemistry, Binghamton University, Binghamton, NY, USA   [2]Department of Pathology and Cell Biology, Columbia University Medical Center, New York, NY, USA

Correspondence: ssolmaz@binghamton.edu

Rab6-positive vesicles via distinct adapters and important for the process, to help the vesicles to reach the freshly polymerized plus-ends of microtubules, to which kinesin-1 binds poorly (Serra-Marques et al, 2020). This is important because the exocytosis hotspots are located near plus-ends of microtubules. Rab6[GTP]-positive secretory vesicles are targeted to exocytic hotspots close to focal adhesion points where the dynamic plus-ends of microtubules are attached to the cell cortex by a complex that contains among other components the Rab6 effector ELKS (named after its high content in the amino acids E, L, K, and S, also known as Rab6-interacting protein 2, R6IP2), and ELKS is important to capture vesicles and promote exocytosis at these hotspots (Nakata et al, 1999; Grigoriev et al, 2007; Patwardhan et al, 2017; Fourriere et al, 2019).

In addition to its role in protein secretion, Rab6/BicD2/dynein also coordinates a retrograde Golgi-to-ER vesicle transport pathway that is independent of the COPI-dependent pathway (White et al, 1999). This pathway has important cellular functions, for example, in signaling and G protein–coupled receptor trafficking. Furthermore, all Golgi enzymes are recycled back to the ER in a Rab6-dependent manner during the mitotic dispersal of the Golgi (Sengupta et al, 2015). Thus, Rab6[GTP]/BicD2 have multiple important roles in the transport of secretory and Golgi-derived vesicles, as well as organization of the Golgi apparatus both in neurons and in regular cells.

In vertebrates, BicD2 also recruits dynein to the nuclear envelope via nuclear pore protein Nup358 and the LINC complex component Nesprin-2, which facilitate two distinct nuclear positioning pathways that are activated during two distinct steps in brain development, the apical nuclear migration in radial glial brain progenitor cells (Nup358/BicD2) and neuronal migration in postmitotic neurons (Nesprin-2/BicD2) (Hu et al, 2013; Gonçalves et al, 2020). These pathways are essential for brain development and important for muscle development. The Rab6[GTP]/BicD2 pathway also impacts brain development, by facilitating the transport of vesicles with protein factors important for brain development, including the CRUMBS complex (Rossor et al, 2020; Brault et al, 2022).

The importance of these BicD2-dependent pathways in brain and muscle development is demonstrated by the fact that *BICD2* mutations cause devastating brain and muscle development diseases, including a subset of cases of spinal muscular atrophy, which is in combination the most common genetic cause of death in infants (Neveling et al, 2013; Oates et al, 2013; Rossor et al, 2020; Yi et al, 2023). Several disease mutations are located in the C-terminal cargo-binding domain of BicD2 (BicD2-CTD) and affect the affinity of BicD2-CTD towards distinct cargoes in a different manner, including Nup358, Nesprin-2, Rab6 (Huynh & Vale, 2017; Yi et al, 2023). The R694C human disease mutation of BicD2 causes a fourfold increase in the affinity that is selective towards Nup358 (Yi et al, 2023) and is associated with defects in neuronal migration. In addition, the BicD2 mutations E774G and R747C/F743I each strongly increased binding of BicD2 to Nesprin-2 but diminished binding to Nup358 (Yi et al, 2023). Interestingly, the E774G mutation also diminishes binding to Rab6, whereas the R747C/F743I mutation does not (Noell et al, 2019; Cui et al, 2020). Both mutants caused defects in inter-kinetic nuclear migration of brain progenitor cells but not in neuronal migration (Yi et al, 2023). The different effects of the point

mutations on the affinity of BicD2 towards different cargoes suggest that Nup358, Nesprin-2, and Rab6 bind to distinct but overlapping sites on BicD2 and compete for binding. To fully understand underlying disease causes structural characterization of distinct BicD2/cargo complexes is necessary.

The affinity of BicD2 to different cargoes is also regulated by cyclin-dependent kinase 1 (Cdk1) and Polo-like kinase 1 (Plk1), which are active in the G2 phase of the cell cycle and promote a switch for BicD2 from preferentially binding to Rab6 during the G1 and S phases to preferentially interacting with Nup358 in the G2 phase (Splinter et al, 2010; Baffet et al, 2015; Gallisà-Suñé et al, 2023; Jimenez et al, 2023 *Preprint*).

The structure of the C-terminal minimal cargo-binding domain of BicD2 has been determined and forms a homodimeric coiled coil (Liu et al, 2013; Terawaki et al, 2015; Noell et al, 2019). A structural basis for recognition of Nup358 by BicD2 was recently established (Gibson et al, 2022). The core binding site of Nup358 to BicD2 is formed by a short cargo-recognition α-helix, which is disordered in Nup358 but becomes α-helical in the complex with BicD2. This α-helix is important for modulation of dynein motility and likely stabilizes BicD2/dynein in the active state.

The structure of a BicD2/cargo complex is not available, and it is unknown how BicD2 recognizes Rab6. The minimal Rab6 binding site was mapped to the C-terminal ~50 residues of BicD2 (Liu et al, 2013; Terawaki et al, 2015), but the binding site of BicD2 on Rab6 has not been identified.

It has previously been established that the active-form Rab6[GTP] has a 10-fold higher activity to BicD2 than Rab6[GDP] (Bergbrede et al, 2009). Structural studies of the GTP- and GDP-bound state of Rab6 were performed, and several conformational changes specific to the GTP-bound state were observed in the Switch 1 and Switch 2 regions (Garcia-Saez et al, 2006). Here, we hypothesize that the BicD2 binding site would be located in these regions that undergo structural changes in the GTP-bound state.

To test this hypothesis, we obtained a structural model for the interaction of Rab6 with BicD2, using structure prediction by AlphaFold2 (Jumper et al, 2021; Evans et al, 2022 *Preprint*), combined with mutagenesis. The binding site of BicD2 spans both the Switch 1 and Switch 2 regions of Rab6, explaining why the GTP-bound state has a higher affinity than the GDP-bound state. Mutations of Rab6[GTP], which abolish binding to BicD2, result in severely reduced motility of Rab6-positive vesicles in cells, highlighting the importance of the interaction between Rab6 and BicD2 for activation of plus end– and minus end–directed motility. Our results establish a structural basis for cargo recognition by BicD2, which facilitates transport pathways that are important for vesicle trafficking and brain development.

## Results

### A model of the Rab6[GTP]/BicD2-CTD complex was obtained by structure prediction with AlphaFold2

To establish a structural basis for cargo recognition by BicD2, we used the software ColabFold (Mirdita et al, 2022), which combines

the homology search of MMseqs2 with AlphaFold2 (Jumper et al, 2021; Evans et al, 2022 Preprint) to predict a structural model of the Rab6$^{GTP}$/BicD2-CTD complex. The complex was previously characterized by size-exclusion chromatography coupled with multi-angle light scattering and shown to form a 2:2 hetero-tetramer (Noell et al, 2018). The structure of the C-terminal cargo-binding domain of BicD2 has been determined; it forms a homodimeric, parallel coiled coil (Liu et al, 2013; Terawaki et al, 2015; Noell et al, 2019). The structure of Rab6 with the ligand GTP bound is also established (Bergbrede et al, 2005; Eathiraj et al, 2005). We used a structure of GTP-bound Rab6a with the Q72L mutation, which locks it in the GTP-bound state, as a template for the AlphaFold2 prediction (PDB ID 2GIL) (Martinez et al, 1994; Matanis et al, 2002; Bergbrede et al, 2005).

The structural model of the Rab6$^{GTP}$/BicD2 complex with the highest predicted local distance difference test (pLDDT) scores, and the lowest predicted aligned error (PAE) is shown in Fig 1A. The remaining predicted models are shown in Fig S1.

The pLDDT is a per-residue confidence metric, which estimates how well the prediction would agree with an experimental structure based on a local distance difference test (Jumper et al, 2021; Tunyasuvunakool et al, 2021), which estimates whether the predicted residue has similar distances to neighboring C-alpha atoms as observed in the experimental structure. The pLDDT in the highest ranked model of the Rab6$^{GTP}$/BicD2-CTD complex is above 85–98% with the exception of a few residues at the N- or C-terminus, indicating high confidence in the structure (Fig 1C and D). The PAE gives a distance error for every residue pair, and estimates the error of the position of residue x in Å if the predicted and actual structures are aligned at residue y. For complexes, it is an important error estimate that assesses the respective positioning of individual subunits. For the most part, the PAE of the highest ranked model of the Rab6$^{GTP}$/BicD2-CTD complex is below 5–10 Å, reflecting a high degree of confidence in the prediction (Fig 1B). Only the N-terminal portion of the BicD2-CTD has a somewhat elevated error, likely reflecting the flexibility of the coiled coil. Notably, the C-terminal part that is bound to Rab6 has a lower PAE.

Furthermore, we also used AlphaFold2 (Jumper et al, 2021) to predict a 2:2 complex of full-length BicD2 and Rab6$^{GTP}$ (Fig S2). The model of the minimal Rab6/BicD2-CTD overlays well with the same domains in the predicted structure of the full-length complex, and these domains have reliable PAE and pLDDT scores. However, the PAE and pLDDT scores of the remaining domains of the full-length complex suggest that the prediction of the remaining structure is not reliable (Fig S2). Thus, we focused on the reliable model of the minimal Rab6$^{GTP}$/BicD2-CTD complex.

The highest ranked model of the minimal Rab6/BicD2-CTD complex is also supported by available biochemical data. In the model, Rab6$^{GTP}$ binds to the ~30 C-terminal residues of human BicD2, which is in line with the previously mapped minimal binding site consisting of residues 755–802 of BicD2 (Liu et al, 2013; Terawaki et al, 2015). Overall, the structures of the BicD2-CTD and Rab6$^{GTP}$ in the complex are very similar to the original structures of the individual proteins (Bergbrede et al, 2005; Noell et al, 2019) (Fig S3A and B), but this is to be expected because AlphaFold2 is trained on the structures in the Protein Data Bank (Jumper et al, 2021).

Before our study, the BicD2 binding site on Rab6 had not been mapped. In our model, BicD2 binds to a site on Rab6 that is formed by a β-strand, a short α-helix, and a coil region (Fig 1A). The GTP is close to the BicD2 binding site but does not engage in the interaction. We analyzed the Rab6$^{GTP}$/BicD2-CTD complex for non-covalent interactions, which are summarized in Fig S3D. The presented contact residues are likely reliable, whereas specific salt bridges or hydrogen bonds are not necessarily expected to be accurately predicted.

Fig S3C shows the electrostatic surface potential of Rab6 and BicD2, which highlights that complementary electrostatic interactions are important for stabilizing the complex. BicD2 residues 774–804 form contacts with Rab6 residues 43–82 (Fig S3D), and all Rab6 residues that make contacts with BicD2 are highly conserved (Fig S3E). Interestingly, the binding site of BicD2 spans two regions of Rab6 that have previously been shown to undergo structural rearrangements in the GTP- versus GDP-bound state: Switch 1, which is located in residues 38–50 of Rab6, and Switch 2, which includes residues 67–87 (Garcia-Saez et al, 2006). These results are in line with the observation that BicD2 binds to Rab6$^{GTP}$ with higher affinity compared with the GDP-bound state (Bergbrede et al, 2009).

Overall, the pLDDT and PAE error plots of the highest ranked model of the Rab6$^{GTP}$/BicD2-CTD complex indicate a high degree of confidence in the model, and it also fits well with biological data such as the previously mapped Rab6 binding site on BicD2, as well as previously identified point mutations that diminish binding of Rab6 to BicD2. Notably, the BicD2 binding site is located in two regions of Rab6 that have been previously shown to undergo conformational changes in the active GTP-bound state, potentially explaining why this state has a higher affinity to BicD2 than the inactive GDP-bound state.

### The binding sites of BicD2 for Nup358 and Rab6 are structurally distinct but overlapping

Nup358 and Rab6$^{GTP}$ compete for binding to BicD2, and Nup358 binds to a larger binding site on BicD2 (residues 724–802) compared with Rab6$^{GTP}$ (residues 755–802), in line with its 20-fold higher affinity (Noell et al, 2018). Yet, the binding mode of these two cargoes to BicD2 is distinct. The core BicD2 binding site of Nup358 is formed by a short cargo-recognition α-helix, which is α-helical in the complex but intrinsically disordered in apo-Nup358 (Gibson et al, 2022). In addition, a short intrinsically disordered region of Nup358 binds to the C-terminal half of the BicD2-CTD in an antiparallel manner (Gibson et al, 2023). In comparison, the BicD2 binding site of Rab6 is formed by a β-strand, an α-helix, and a coil region (Fig 1E and F). Rab6 binds to the same region of BicD2 as the intrinsically disordered N-terminal residues of Nup358-min, whereas the cargo-recognition α-helix binds on the center of the BicD2-CTD. Thus, the intrinsically disordered domain of Nup358 is the one that competes for binding to BicD2 with Rab6$^{GTP}$ (Fig 1E and F).

It has previously been shown that the E774A mutation reduces binding to Rab6$^{GTP}$ and Nup358, whereas the F743I/R747C mutation does not impact Rab6 binding but diminishes Nup358 binding (Noell et al, 2019; Cui et al, 2020; Yi et al, 2023). Our model of the

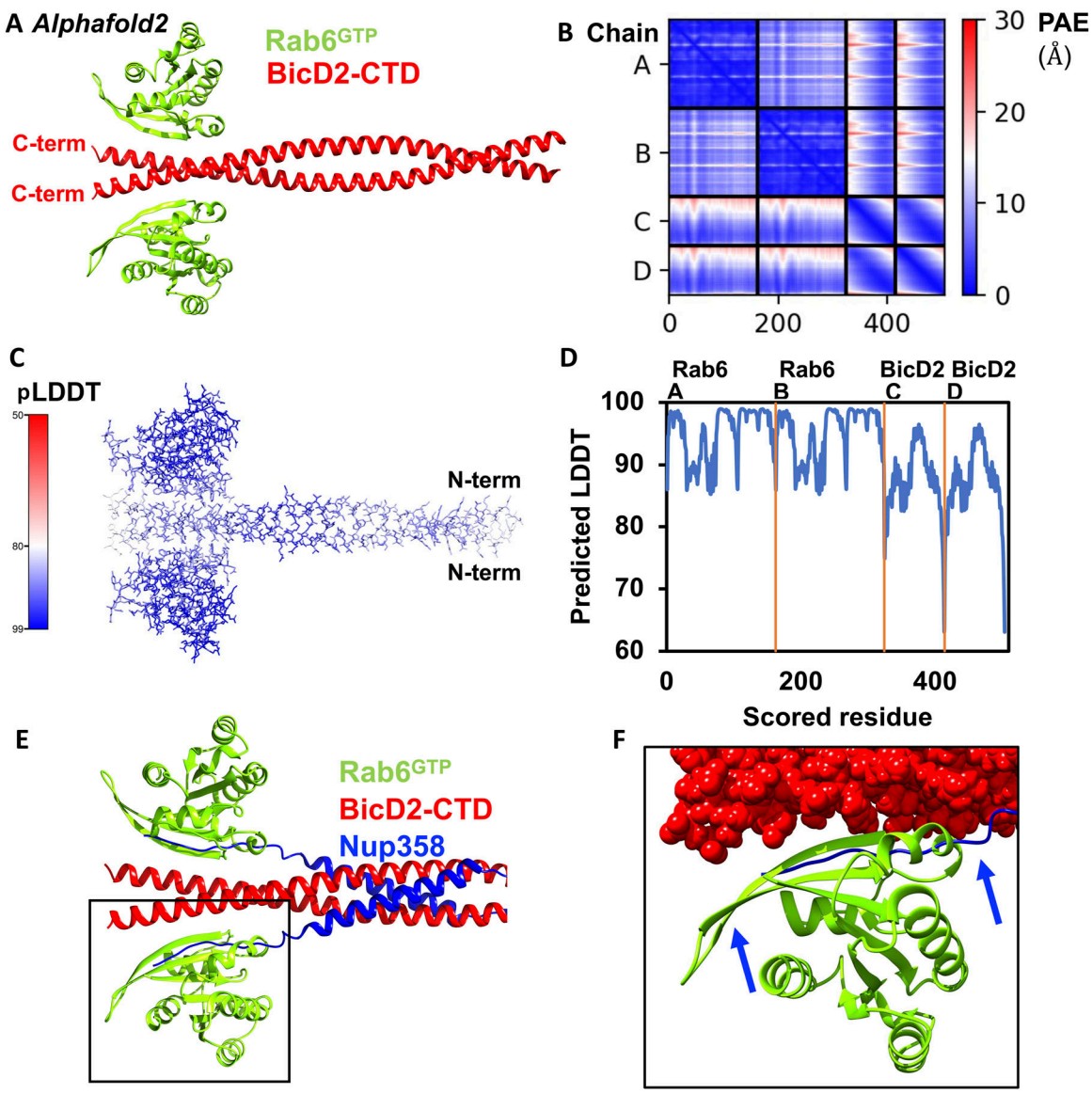

**Figure 1. A structural model of the Rab6$^{GTP}$/BicD2-CTD complex was obtained from AlphaFold2.**
**(A)** Cartoon representation of the highest ranked structural model of the Rab6$^{GTP}$/BicD2-CTD complex. **(B)** Associated predicted aligned error plot, in which each residue in the structure is plotted on the x-axis and y-axis. The plot is colored by a gradient that indicates the predicted aligned error estimate in Å for each residue pair (blue: low error; white: acceptable threshold; red: high error). Chain A, B; Rab6; Chain C, D: BicD2. **(C, D)** Associated per-residue local distance difference test (pLDDT) confidence scores. **(C)** The structural model is colored by a gradient that indicates the pLDDT confidence score (100–90, blue: high confidence; 80, white: confident; 50, red: low confidence). **(D)** pLDDT plot. Four lower ranked structural models and their error plots are shown in Fig S1A–C. **(E)** Least-squares superimposition of the structural models of the Rab6$^{GTP}$/BicD2-CTD complex with the Nup358-min/BicD2-CTD complex (Gibson et al, 2023) from AlphaFold2. The BicD2 chains of the Nup358-min/BicD2-CTD complex are omitted for clarity. The BicD2 binding site is boxed. **(E, F)** The boxed area from (E) is enlarged, highlighting the binding site of the N-terminal intrinsically disordered region of Nup358-min (Gibson et al, 2023), which binds to the same binding site on BicD2-CTD as Rab6$^{GTP}$. BicD2 is shown in sphere representation. Note that Rab6 and Nup358 have distinct binding modes on BicD2, but the binding sites overlap, explaining why these cargoes compete.
Source data are available for this figure.

Rab6$^{GTP}$/BicD2-CTD complex is in line with these results, because F743 and R747 are N-terminal of the Rab6$^{GTP}$ binding site, whereas E774 is a contact residue with Rab6$^{GTP}$.

To conclude, the Nup358 and Rab6$^{GTP}$ binding sites on BicD2 are overlapping but structurally distinct, in line with the observation that BicD2 disease mutations affect affinities towards these cargoes differently.

## The binding sites of Rab6$^{GTP}$ for Rab6 and ELKS are overlapping, providing a mechanism for competition

ELKS (also known as Rab6IP2, CAST, or ERC) is a Rab6$^{GTP}$ interactor that captures Rab6-coated vesicles near plus-ends of microtubules at exocytosis hotspots in neurons and other cells and is a key regulator for protein secretion (Grigoriev et al, 2007; Patwardhan

et al, 2017; Fourriere et al, 2019). A structure of a minimal Rab6$^{GTP}$/ELKS complex has been previously established, and it was observed that BicD2 and ELKS do not co-localize in cells, indicating that they compete for binding (Jin et al, 2023). Although ELKS has a lower affinity to Rab6 than BicD2, ELKS may form liquid–liquid phase separation condensates that increase its local concentration, are able to capture Rab6 vesicles, and thereby compete with other Rab6 effectors (Jin et al, 2023). However, the molecular basis for competition of BicD2 and ELKS for Rab6 binding was unknown. Here, a least-squares superimposition of the minimal Rab6$^{GTP}$/ELKS complex (Jin et al, 2023) with our structural model of the Rab6$^{GTP}$/BicD2-CTD complex suggests that both effectors bind to distinct but overlapping binding sites on Rab6$^{GTP}$. An analysis of the interface confirms that several Rab6$^{GTP}$ contact residues engage in binding to both BicD2-CTD and ELKS (Fig S4A–C, Table S1). Competition of the Rab6 effectors BicD2 and ELKS could be an efficient means of organizing a cascade of alternating protein interactions that enable Rab6 to regulate protein trafficking and secretion.

Rab6$^{GTP}$ has multiple other interactors, and we investigated whether they share the same binding site. Least-squares superimpositions of the predicted Rab6$^{GTP}$/BicD2-CTD complex with three other structures from the PDB: Rab6 bound to Kif20A (Fig S5A) (Miserey-Lenkei et al, 2017), GCC185 (Fig S5B) (Burguete et al, 2008), or R6IP1 (Fig S5C and D) (Recacha et al, 2009), show that each of these proteins binds to distinct but overlapping binding sites on Rab6; therefore, it is likely that BicD2 competes with these three other interactors for binding to Rab6 (Fig S5).

Rab6$^{GTP}$ also can bind directly to residues 737–916 of the dynactin-interacting domain p150$_{glued}$ (Bergbrede et al, 2009). Because a structure of the complex is not available, we used AlphaFold2, and the highest ranked prediction of a Rab6$^{GTP}$/p150$_{glued}$ complex with 2:2 stoichiometry received pLDDT scores mostly above 80 and PAE scores below 10 Å, indicating that the prediction is likely reliable (Fig S6). A least-squares superimposition of the predicted structure of the minimal Rab6$^{GTP}$/p150$_{glued}$ complex with the Rab6$^{GTP}$/BicD2-CTD complex is shown in Fig S6, which suggests that p150$_{glued}$ and BicD2-CTD compete for binding to Rab6$^{GTP}$. However, this remains to be experimentally confirmed.

To conclude, our data suggest that all Rab6 interactors analyzed here, including BicD2, ELKS, the dynactin subunit p150$_{glued}$, Kif20A, GCC185, and R6IP1, bind to distinct but overlapping binding sites on Rab6$^{GTP}$ and compete for binding. Thus, these Rab6 interactions are likely organized as a cascade and not formed simultaneously.

### The structural model of the Rab6$^{GTP}$/BicD2-CTD complex is validated by mutagenesis

Next, we validated the BicD2 binding site on Rab6$^{GTP}$ by mutagenesis. The contact residues of Rab6$^{GTP}$ from the AlphaFold2 model (Fig S3D) were mutated to alanine along with the remaining Rab6 residues of the Switch 1 and Switch 2 regions, which undergo structural changes in the GTP-bound state (Fig 2A). For these experiments, the Q72L mutant of Rab6 was used as WT; this mutant is locked in the GTP-bound state that has a higher affinity to BicD2-CTD.

To identify residues that are important for binding of Rab6 to BicD2, binding of BicD2-CTD to GST-tagged Rab6$^{GTP}$ mutants was

assessed by pull-down assays. The elution fractions were analyzed on SDS–PAGE, and the intensities of the gel bands were quantified (Fig 2B and C; a representative full dataset is shown in Fig S7). These experiments revealed several Rab6 residues that are important for binding to BicD2: five residues in the Switch 1 region of Rab6, eight residues in the Switch 2 region of Rab6, and five residues in the connecting interswitch region (red in Fig 2C). It should be noted that three of the Rab6 residues that are contact residues in the AlphaFold2 model and that are also confirmed to be important for binding to BicD2 constitute the invariant hydrophobic triad, which is conserved in the Rab family of proteins and forms a hydrophobic switch region interface: residues F50, W67, and Y82. This invariant hydrophobic triad has been shown to undergo structural changes upon activation in the GTP-bound state that are a determinant for effector recognition, and specifically for BicD2, as we show here (Dumas et al, 1999; Ostermeier & Brunger, 1999; Merithew et al, 2001). Notably, the residues established by mutagenesis that are essential for the Rab6$^{GTP}$/BicD2 interaction validate the structural model of the Rab6$^{GTP}$/BicD2-CTD complex (Figs 2 and S3, Table S4).

Fig 2C shows the sequence of Rab6, in which the mutated residues that showed reduced binding in the pull-down assays are colored red. In addition, Rab6 residues that form contacts with both GTP and Mg$^{2+}$ are indicated by an asterisk and Rab6 residues that form contacts with GTP are indicated by a plus sign (see Table S2). It should be noted that mutations of GTP- or Mg$^{2+}$-contacting residues may possibly alter the interaction of Rab6 with GTP, which would in turn impact binding to BicD2.

To exclude that the mutations that decreased binding to BicD2 resulted in misfolding of Rab6$^{GTP}$, which could result in lowered binding even if the mutated residues were not engaged in the interaction with BicD2, we characterized the secondary structure of the Rab6$^{GTP}$ mutants by circular dichroism (CD) spectroscopy.

Representative CD wavelength scans of three mutants overlaid with the WT spectra are shown in Fig 2, and the full dataset of all mutants is shown in Fig S8. Two local minima are observed in the spectra at 208 and 222 nm, which are characteristic of α-helical structures. The CD wavelength scan of the mutants K53A, I79A, and Y42A is very similar to the WT, confirming that the mutation does not result in misfolding or large structural changes compared with the WT (Fig 2D). In total, the secondary structure content of all 19 mutants that reduced binding to BicD2 (red in Fig 2C) was assessed by CD spectroscopy. The CD spectra of all 19 mutants resembled the WT spectra, suggesting that all mutants were correctly folded and did not have significant structural changes compared with the WT (Fig S8). The secondary structure content of all CD spectra was estimated with the program BeStSel and is summarized in Table S3. It should be noted that some minor changes were observed in some of the CD spectra of the mutants, as well as in the derived secondary structure content (Fig S8, Table S3). These differences are not significant, because the experimental error of the molar ellipticity is 3.5–5%. We recently determined this error by calculating the SD of the molar ellipticity at 208 and 222 nm from 10 experiments for which the samples were independently prepared (including the determination of the protein concentration, which is the main source of the experimental error) (Cui et al, 2020).

Fig 3A and B shows a close-up of the structure of the Rab6$^{GTP}$/BicD2-CTD complex with the confirmed Rab6 residues that are

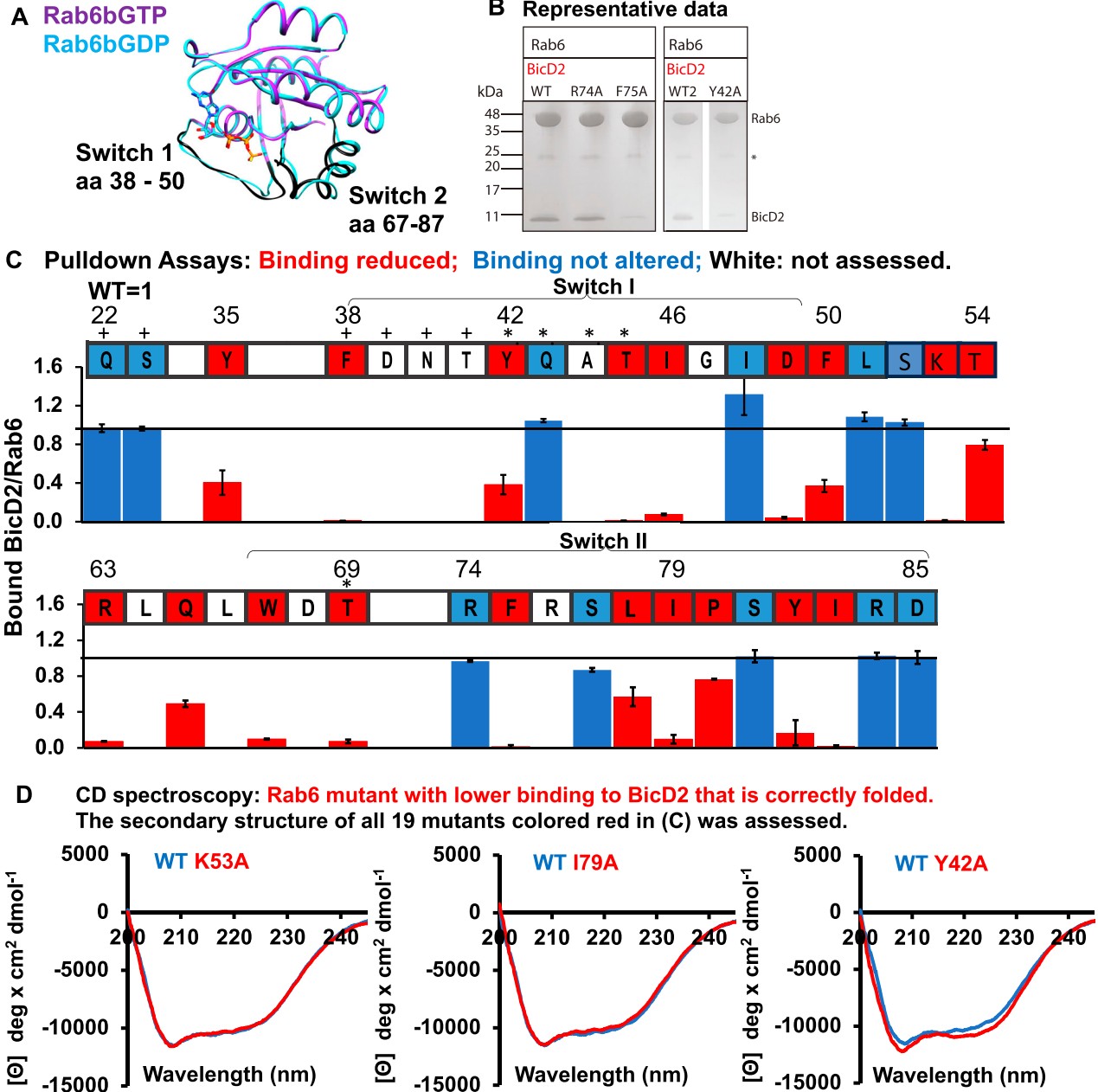

**Figure 2.  The structural model of the Rab6$^{GTP}$/BicD2-CTD complex is validated by mutagenesis.**
**(A)** Least-squares superimposition of the structures of GTP (purple and black)- and GDP-bound (cyan) Rab6b in cartoon representation (Garcia-Saez et al, 2006). Switch 1 and Switch 2, two regions that undergo structural changes in the GTP-bound state, are colored black in Rab6b$^{GTP}$ (Garcia-Saez et al, 2006). **(B, C)** All BicD2-contacting Rab6$^{GTP}$ residues from the AlphaFold2 model and additional residues from the Switch 1 and Switch 2 regions were mutated to alanine, and binding to BicD2-CTD was assessed by pull-down assays. The elution fractions were analyzed on SDS–PAGE, and the intensities of the gel bands were quantified with ImageJ (Schneider et al, 2012). **(B)** Representative SDS–PAGE of elution fractions of GST pull-downs. Left panel: WT, R74A, and F75A mutant. Right panel: Second WT sample and Y42A mutant. Molar masses of standards are indicated on the left. A full representative dataset is shown in Fig S7. Three datasets were collected. An asterisk indicates the position of GST. **(C)** Sequence of Rab6 is shown, and residues for which the mutations reduced binding to BicD2 are colored red, residues for which the mutations did not diminish binding are colored blue, and residues that were not assessed are colored white. Rab6 residues that form contacts with both GTP and Mg$^{2+}$ are indicated by an asterisk, and Rab6 residues that form contacts with GTP are indicated by a plus sign (see Table S2). Middle panel: bar graph showing the ratio of bound BicD2/Rab6$^{GTP}$ from pull-down assays normalized respective to the WT (WT = 1; reduced binding = red, normal binding = blue). Ratios were averaged from three experiments, and the error bars show the SD. **(C, D)** Rab6 mutants that resulted in reduced binding to BicD2 (colored red in (C)) were characterized by circular dichroism (CD) spectroscopy to assess their secondary structure content. **(C)** Residues colored white and blue in (C) were not assessed. Representative CD wavelength scans are shown for the WT (blue) and three mutants (red; K53A, I79A, and Y42A). CD wavelength scans for all mutants are shown in Fig S8. Note that the CD spectra of all 19 mutants that reduced binding were similar to the WT spectra, suggesting that they do not misfold. All CD experiments were performed three times with independently purified samples. See also Tables S3 and S4. Source data are available for this figure.

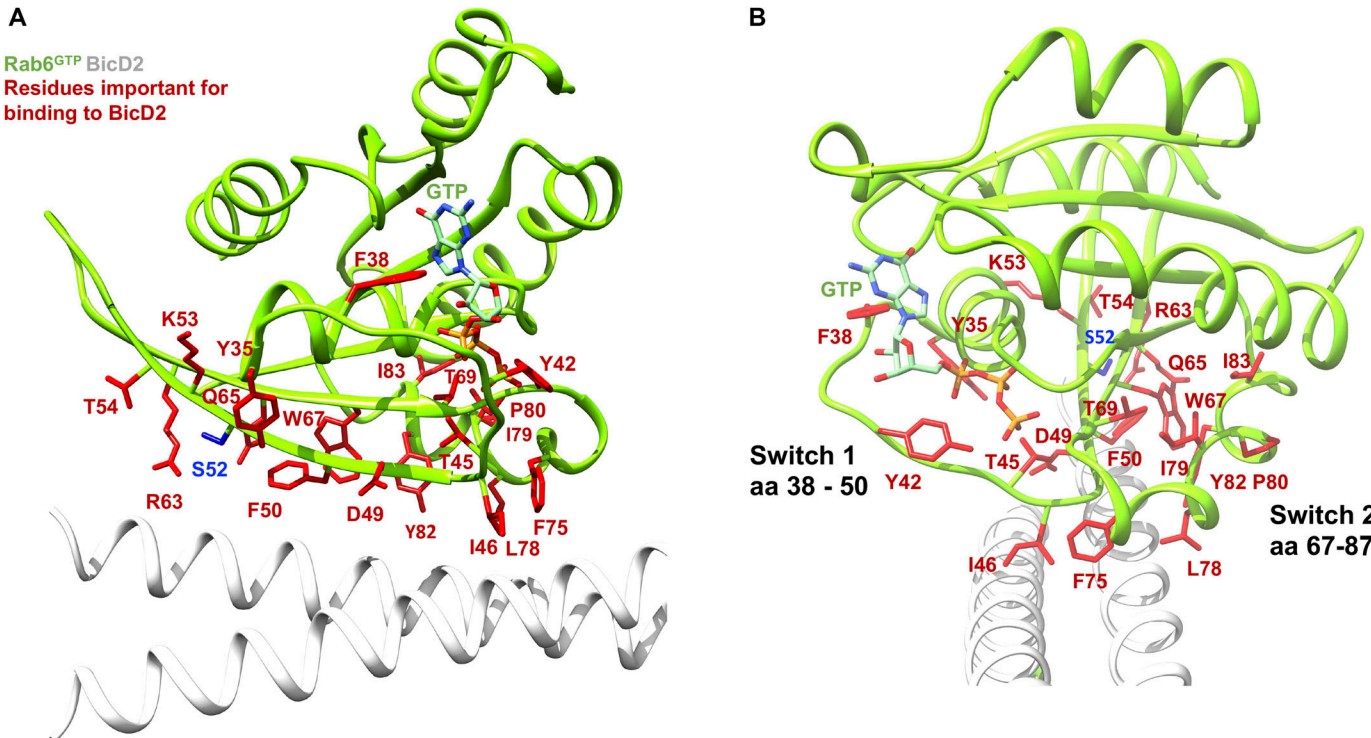

**Figure 3. Interactions between Rab6^GTP and BicD2-CTD are mediated by aromatic and hydrophobic residues.**
**(A, B)** Cartoon representation of the structure of the Rab6^GTP/BicD2-CTD complex (Rab6 green, BicD2-CTD white) shown rotated by 90°. GTP is shown in stick representation. The residues from Fig 2, which result in reduced binding after mutagenesis, are shown in red stick representation. S52, which is phosphorylated in the G2 phase, is shown in blue.

essential for the interaction with BicD2 in red stick representation. Notably, the interacting residues mainly consist of aromatic residues, as well as some hydrophobic residues: F38, Y42, I46, F50, W67, F75, I79, Y82, and I83, and the positively charged R63. These residues project from the surface of Rab6-like fingers. It is likely that the interaction is mainly stabilized by hydrophobic interactions. In addition, the interaction is further strengthened by complementary charged surfaces (Fig S3C).

Furthermore, several residues of BicD2 were previously identified that are important for binding to Rab6^GTP: E774, L782, R783, M784, I786, K789, L790 (Liu et al, 2013; Terawaki et al, 2015). All seven residues form non-covalent interactions with Rab6^GTP in our structural model of the complex (Fig S3D, Table S4). Mutations of BicD2 residues 745, 750, 755, or 756 to alanine, which are N-terminal of the minimal binding site, do not affect binding to Rab6^GTP, as to be expected (Gibson et al, 2023).

It has been established that Rab6 is phosphorylated at position S52 in the G2 phase of the cell cycle by Polo-like kinase 1 (Plk1), which weakens the interaction with BicD2 (Jimenez et al, 2023 *Preprint*). A phosphomimetic S52D mutation had a similar effect, whereas a phosphonegative mutation S52A had no effect on the interaction (as observed in our pull-down assays as well, Fig 2). Notably, S52 is mediating a contact with BicD2. The close-by residues R63 (positively charged) and Q65 (polar) are important for the interaction between BicD2 and Rab6 and could potentially form a salt bridge and a hydrogen bond with negatively charged residues E797 and E782 of BicD2 (Fig S9A and B). It is conceivable that a phosphorylated S52 residue would form an intramolecular salt bridge with R63 and a hydrogen bond with Q65, which could weaken the interaction overall and there is expected to be like-charge repulsion between phosphorylated S52 of Rab6 and E797 and E783 of BicD2.

To conclude, our results from mutagenesis and CD spectroscopy confirm our structural model of the Rab6^GTP/BicD2 complex.

### Structural basis for the higher affinity of GTP-bound Rab6 to BicD2

To understand the increased affinity of the GTP-bound state of Rab6 towards BicD2 (Bergbrede et al, 2009), we compared the structures of GTP- and GDP-bound apo-Rab6b, which is a close homolog to Rab6a with 94% sequence identity. Structural changes are observed in the Switch 1 and Switch 2 regions of Rab6b^GTP; these regions are located close to the GTP ligand (black in Fig 2A) (Garcia-Saez et al, 2006).

The structure of Rab6^GTP in the complex with the BicD2-CTD is very similar to the structure of apo-Rab6b^GTP (and also apo-Rab6a^GTP, Fig S3B), and no significant structural changes are observed in the protein backbone or the interacting residues (Fig 4B).

Fig 4A shows a least-squares superimposition of the structures of the Rab6^GTP/BicD2-CTD complex with the structure of apo-Rab6b^GDP (Garcia-Saez et al, 2006). Several GTP-dependent structural changes are observed: the loop of Switch 1 and the α-helix of

**A**

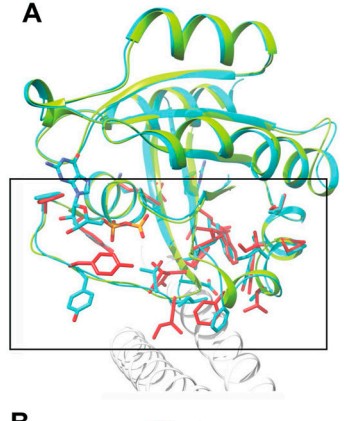

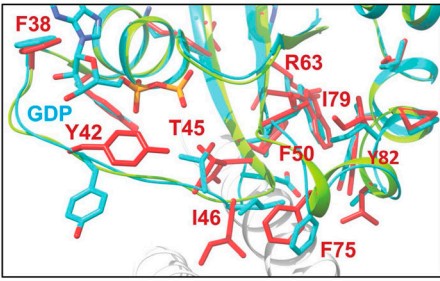

**B**

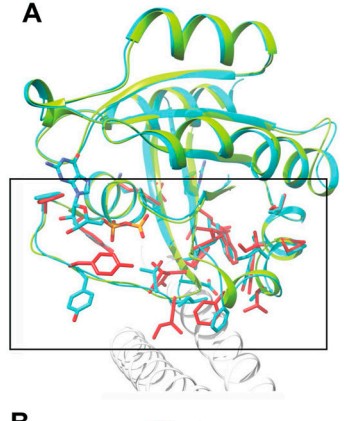

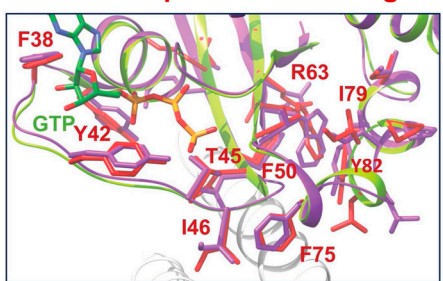

**Figure 4. Structural basis for the increased affinity of GTP-bound Rab6 to BicD2.**
**(A)** Least-squares superimposition of the structure of the Rab6$^{GTP}$/BicD2-CTD complex (Rab6: green; BicD2: gray; interface residues: red) and the structure of Rab6b$^{GDP}$ (cyan) (Garcia-Saez et al, 2006) in cartoon representation. The Rab6 residues that are important for the interaction with BicD2 (Fig 2C) are shown in cyan (Rab6b$^{GDP}$) and red (Rab6$^{GTP}$/BicD2-CTD complex) stick representation. The boxed area is shown enlarged on the right. Note that several of the residues that are important for the interaction between Rab6 and BicD2 undergo conformational changes in the GTP-bound state, likely explaining the higher affinity of active Rab6$^{GTP}$ to BicD2, as it allows the formation of additional hydrophobic interactions. **(B)** Least-squares superimposition of the structures of the Rab6$^{GTP}$/BicD2 complex-CTD (Rab6: green; BicD2: white; residues important for binding: red) and Rab6$^{GTP}$ (purple) (Bergbrede et al, 2005) in cartoon representation. The Rab6 residues that are important for the interaction with BicD2 (Fig 2C) are shown in red (complex) and purple (Rab6$^{GTP}$) stick representation. The boxed area is shown enlarged on the right.

Switch 2 shift their position and undergo structural rearrangements. Notably, the structure of several BicD2 binding residues (colored red) is rearranged. Large-scale structural changes are observed for the hydrophobic and aromatic residues Y42, I46, and F75 and for the positively charged residue R63, which are essential for binding to BicD2. Several other residues that are necessary for the interaction with BicD2 undergo more subtle structural rearrangements: F50, I79, and Y82 (Fig 4A).

We conclude that in the GTP-bound state of Rab6, several aromatic and hydrophobic residues are repositioned compared with the GDP-bound state. These structural rearrangements allow these residues to form a larger number of hydrophobic interactions with BicD2 in the GTP-bound state, thereby increasing the affinity. In the GDP-bound state, these residues are positioned away from the binding site, lowering the binding affinity to BicD2.

### Point mutations that disrupt the Rab6/BicD2 interaction strongly diminish the motility of Rab6-positive vesicles and decrease co-migration of these vesicles with BicD2

Because we have identified several point mutations of Rab6, which selectively disrupt the interaction with BicD2, we investigated how these mutations modify the motility of Rab6-positive vesicles and the co-migration of Rab6 with BicD2 through live imaging in HeLa cells.

For the in vitro binding assays above, the Q72L mutant of Rab6 was used, which locks it in the GTP-bound state, as it has a higher

affinity towards BicD2. It should be noted that for the live-cell imaging, WT Rab6 was used, which did not carry this mutation. GTP-bound Rab6 is the active form that is anchored in membranes by prenylation, whereas the inactive Rab6$^{GDP}$ state is released from the membrane into the cytosol by GDI (Goud et al, 1990; Martinez et al, 1994; Grigoriev et al, 2007) (see Fig S10). Thus, Rab6 that localizes to vesicle membranes is expected to be predominantly bound to GTP.

To assess how the mutations modify the motility of Rab6-positive vesicles, we transiently transfected mCherry-tagged Rab6 into HeLa cells and performed live-cell imaging. Fig 5A shows several time points and a kymograph for mCherry (mCh)-Rab6-WT–positive vesicles, which are derived from Video 1. Single vesicles are highlighted by distinctly colored arrows, which suggest normal motility of Rab6-positive vesicles in the WT (Fig 5A, Video 1). Notably, the F38A and Y42A mutations, which selectively disrupt the Rab6/BicD2 interaction in vitro, result in a strong reduction in the motility of the Rab6-positive vesicles (Fig 5B–D and Video 2 and Video 3). To quantify the effect of these mutants on the motility of the Rab6-positive vesicles, we determined the number of trajectories of Rab6-positive vesicles for the WT, the F38A mutant, and the Y42A mutant using automated particle tracking that identifies the trajectories of moving particles (Figs 5D and S11A–D, Video 4 and Video 5). A recent study concluded that the results obtained from this automated analysis were very similar to those determined manually from the analysis of kymographs drawn along Rab6-positive tracks (Grigoriev et al, 2007; Schlager et al, 2014b; Serra-Marques

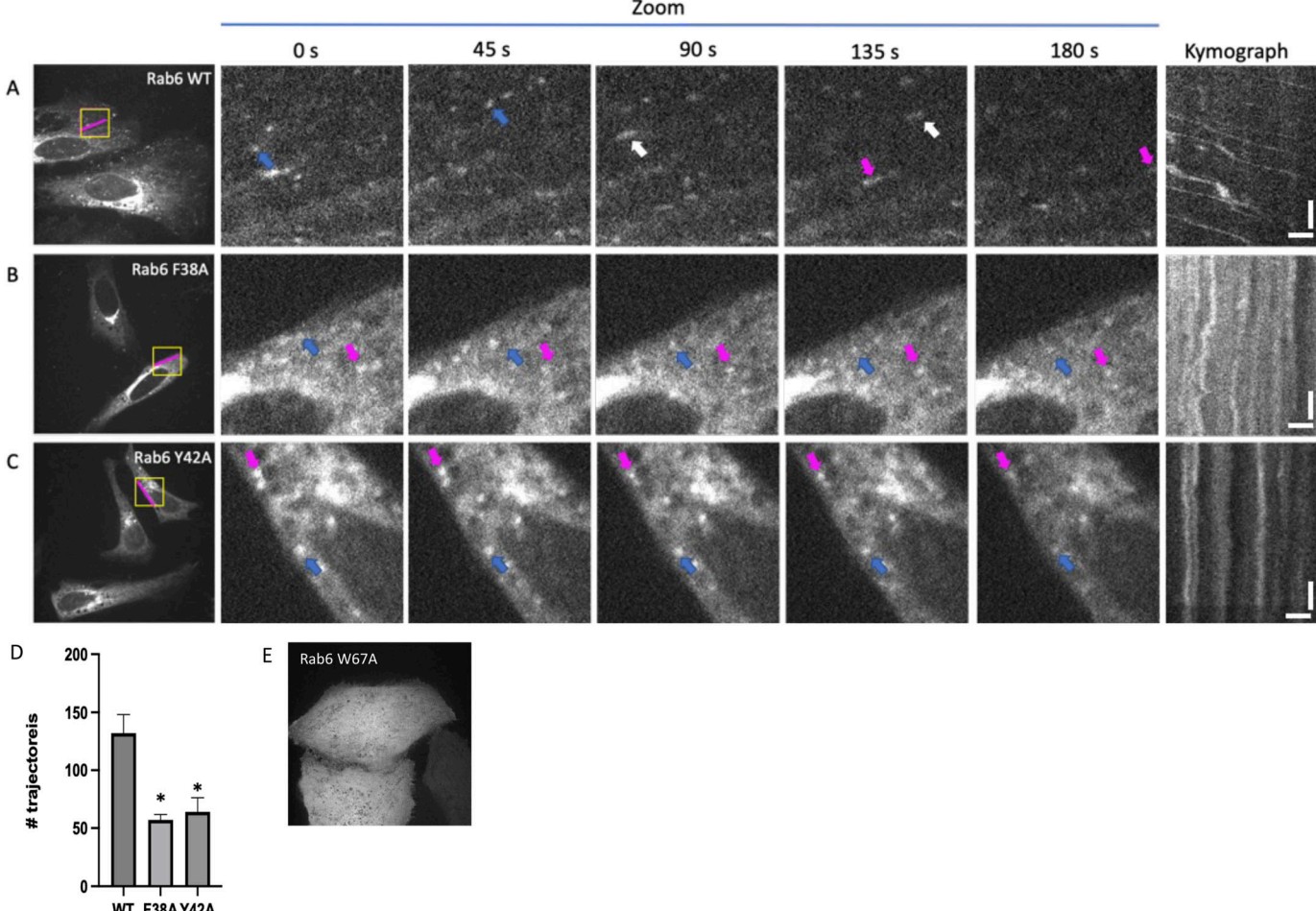

**Figure 5. Point mutations that disrupt the Rab6/BicD2 interaction strongly decrease the motility of Rab6-positive vesicles in cells.**
**(A, B, C)** Live-cell imaging micrographs of (A) mCherry-Rab6-WT, (B) mCherry-Rab6/F38A, and (C) mCherry-Rab6/Y42A expressed in HeLa cells derived from Video 1, Video 2, and Video 3. Different time points for Rab6-positive vesicles are shown for a zoomed area highlighted with a yellow square. Arrows with distinct colors show the migration of single vesicles. Kymograph (far right) show the migration of vesicles in the 15-μm segment highlighted with a magenta line. The kymograph scale bars on the x-axis measure 2.5 μm, and the scale bars on the y-axis measure 30 s. **(D)** Quantification of the number of Rab6-positive vesicle trajectories identified during live imaging in cells expressing fluorescent fusion proteins of Rab6-WT, Rab6/F38A, and Rab6/Y42A. Both mutants show a significant decrease in moving Rab6-positive vesicle trajectories compared with the WT. *P-values of 0.0090 for WT versus F38A, and 0.0113 for WT versus Y42A were obtained with the Kruskal–Wallis test (n = 3 independent experiments) (see also Fig S11A–D). **(E)** Micrograph of immunostained fixed HeLa cells that transiently express a red fluorescent mCherry fusion protein of Rab6/W67A. The Rab6/W67A mutant disperses into the cell.
Source data are available for this figure.

et al, 2020). The results of our analysis showed that the number of motile vesicle trajectories is significantly reduced in both the F38A and the Y42A mutants compared with the WT (Fig 5D).

Although the number of the motile Rab6-positive vesicles is greatly reduced for the mutants, the Rab6-related fluorescence signal in the Golgi area remains strong, suggesting that these mutations do not affect localization of Rab6 to the Golgi membranes. Because membrane-associated Rab6 is mainly in the GTP-bound and active state, the strong localization of the mutants and the WT to Golgi-derived Rab6-positive vesicles suggests that the F38A and Y42A mutations do not affect the equilibrium between GDP- and GTP-bound states of Rab6 and also do not compromise activation and membrane integration of Rab6. In comparison, Fig S10A and C shows micrographs of cells expressing fluorescent fusion proteins of WT Rab6 and the Rab6/Q72L mutant that is

locked in the GTP-bound state, and for both conditions, Rab6 localizes robustly to Golgi-derived vesicles, somewhat similar as observed for the F38A and Y42A mutants. In comparison, the Rab6/T27N mutant, which is locked in the GDP-bound state, disperses into the cell, resulting in an easily detectable phenotype (Fig S10B). It should be noted that one of the mutants we tested, Rab6/W67A, which is a contact residue in the AlphaFold2 model, also disperses into the cell, indicating that the mutation could affect either binding to GTP or activation of Rab6, which would subsequently impact membrane integration (Fig 5E). This is to be expected, as W67 is a member of the invariant hydrophobic triad that is conserved in Rab proteins. This triad is involved in the activation mechanism of Rab6, and important for recognition of effectors such as BicD2 (Dumas et al, 1999; Ostermeier & Brunger, 1999; Merithew et al, 2001). We conclude that the Y42A and F38A mutations do not significantly

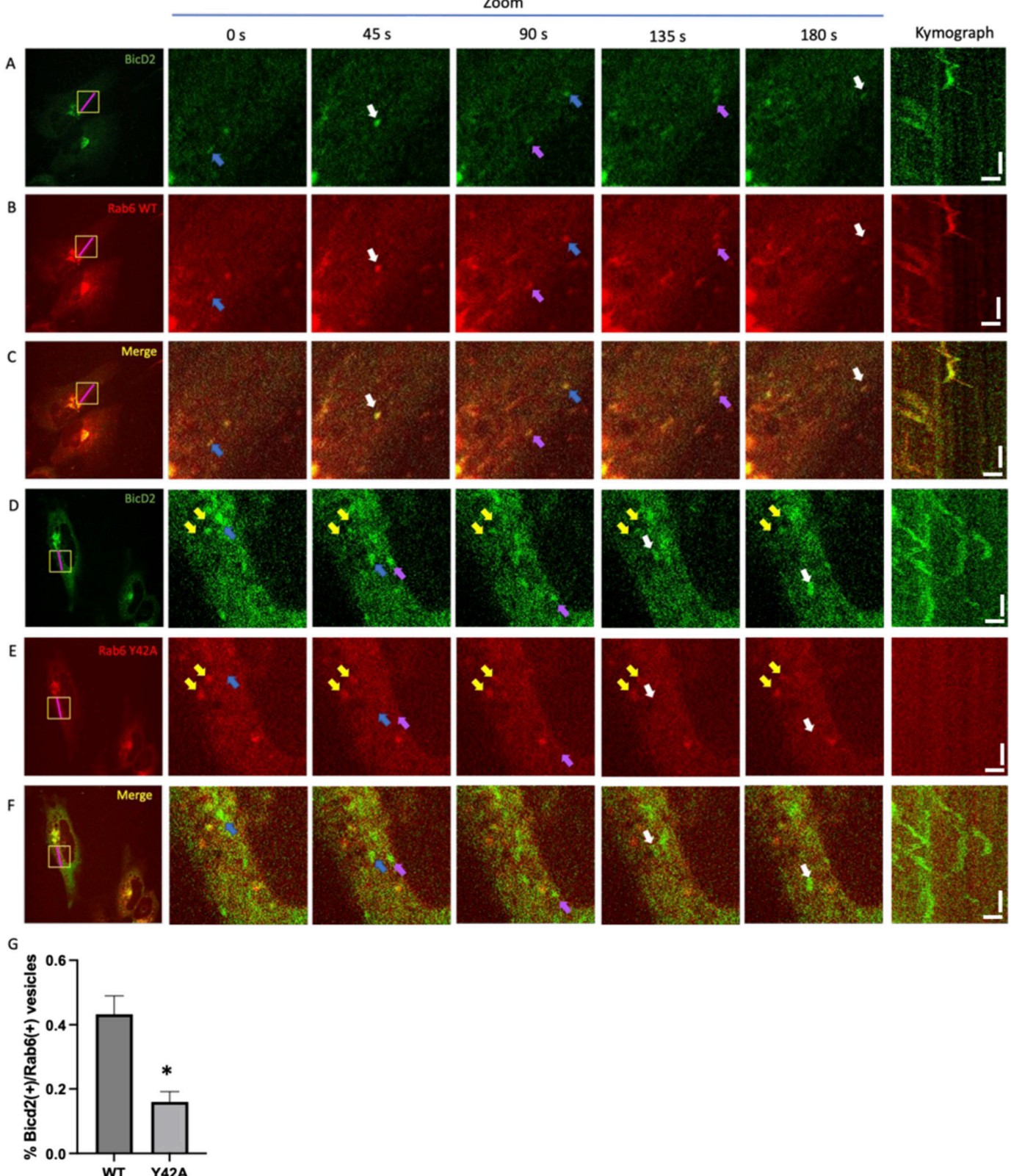

diminish binding of GTP or activation of Rab6, as it would result in their dispersal from the Golgi membranes.

We also quantified co-migration of Rab6- and BicD2-positive vesicles to assess whether the observed disruption of the Rab6/BicD2 interaction by the mutations results in reduced co-migration in the context of live cells. BicD2-GFP and Rab6-mCherry were transiently co-expressed, and the cells were monitored by live-cell imaging. For WT Rab6 and BicD2, Rab6-positive vesicles display normal motility and Rab6 and BicD2 co-localize throughout the experiment, suggesting that they co-migrate as expected (see Video 6, Video 7, and Video 8 and the derived time points and kymograph shown in Fig 6A–C). We can see in the movies, kymographs, and images at distinct time points that Rab6/Y42A-positive vesicles are negative for BicD2 and virtually immotile (yellow arrows, Fig 6D–F, Video 9, Video 10, and Video 11). BicD2-positive/Rab6-negative vesicles, however, display normal motility (blue, purple, and white arrows, Fig 6D–F, Video 9, Video 10, and Video 11). These results suggest decreased Rab6/BicD2 co-migration of the Rab6/Y42A mutant compared with the WT. We also quantified the ratio of co-localization of Rab6 and BicD2, which is significantly decreased in the Y42A mutant compared with the WT (Fig 6G). F38A has a similar effect to Y42A (Video 12).

These experiments confirm that Y42 is a key residue for stabilizing the Rab6/BicD2 interaction in cells and therefore support our model of the Rab6/BicD2 complex, as well as the Rab6 residues that are important for the interaction with BicD2.

To conclude, our data suggest that point mutations that selectively disrupt the interaction between Rab6 and BicD2 in our in vitro binding assays result in severely impacted motility of Rab6-positive vesicles in cells. In addition, these mutations also disrupt co-localization and co-migration of Rab6 and BicD2 in cells, suggesting that they are key residues for stabilizing the Rab6/BicD2 interaction, thereby supporting our model of the Rab6/BicD2 complex.

## Discussion

Rab6 is a key regulator of protein secretion and serves as an identity marker for secretory and Golgi-derived vesicles. Here, we established a structural model of the $Rab6^{GTP}$/BicD2 complex, using AlphaFold2 (Jumper et al, 2021), mutagenesis, and CD spectroscopy. The Rab6 and Nup358 binding sites on BicD2 partially overlap (Gibson et al, 2022, 2023), explaining why these two cargoes compete for binding. The BicD2 binding site of Rab6 is formed by a $\beta$-strand, an $\alpha$-helix, and a coil region and thus different from the cargo-recognition alpha-helix of Nup358 (Gibson et al, 2022, 2023). The binding site spans two regions of $Rab6^{GTP}$, which are known to undergo structural changes in the GTP-

bound compared with the GDP-bound state (Garcia-Saez et al, 2006). Several hydrophobic residues and a charged residue of Rab6 are repositioned for enhanced interactions with BicD2 during the transition to the GTP-bound state, explaining why it has a higher affinity than the GDP-bound state. Several mutants that disrupt the interaction between Rab6 and BicD2 result in loss of co-localization and severely reduced motility of Rab6-positive vesicles in cells, suggesting that the interaction between Rab6 and BicD2 is crucially important for activation of motility of the multi-motor complex that also includes dynein and kinesin-1. Our results expand our understanding of the key role of the Rab6/BicD2 interaction in protein secretion and provide new insights into how BicD2 selects its cargo for transport.

We previously established that BicD2 recognizes its cargo Nup358 by a short cargo-recognition alpha-helix, which is intrinsically disordered in apo-Nup358 but alpha-helical in the complex with BicD2 (Gibson et al, 2022, 2023). The BicD2 binding site of $Rab6^{GTP}$ is structurally distinct and formed by a beta-strand, an alpha-helix, and a coil region that spans Switch 1 and 2, two regions that are known to undergo structural changes in the GTP-bound state. It should be noted that the X-ray structures of the individual proteins Rab6 and BicD2-CTD are very similar to the structures of these proteins in the complex, and we cannot fully exclude that structural changes occur during complex formation. However, in support of our results, the published analysis of the transient binding kinetics of Rab6 to BicD2 by fluorescent stopped-flow technology suggests that the interaction is best described by a single-step mechanism and does not appear to involve large structural rearrangements in $Rab6^{GTP}$ or the BicD2-CTD when the complex is formed (Bergbrede et al, 2009).

We have characterized here the effect of mutations that disrupt the interaction between Rab6 and BicD2 in cells. It should be noted that the mutations likely do not affect the equilibrium between GDP- and GTP-bound states of Rab6. Membrane-associated Rab6 exists mainly in the GTP-bound state, whereas GDP-bound Rab6 is released to the cytosol by GDI, and the mutations do not impact the localization of Rab6 at the Golgi membranes.

WT $Rab6^{GTP}$ and BicD2 co-localize in cells, and we show here that mutants of $Rab6^{GTP}$, which disrupt binding to BicD2, result in a loss of co-localization, confirming our key residues for the Rab6 interaction in the context of cells. Notably, for the assessed mutants that disrupt the interaction, the motility of Rab6-positive vesicles is much reduced compared with WT Rab6, suggesting that the interaction between $Rab6^{GTP}$ and BicD2 is important for activation of motility of the multi-motor complex, which includes apart from $Rab6^{GTP}$/BicD2 also dynein/dynactin and kinesin-1 (Grigoriev et al, 2007; Serra-Marques et al, 2020). For the retrograde transport from

**Figure 6. Point mutations that disrupt the Rab6/BicD2 interaction strongly diminish the motility of Rab6-positive vesicles and decrease co-migration of these vesicles with BicD2.**
**(A, B, C, D, E, F)** Live-cell imaging micrographs of mCherry-Rab6-WT/BicD2-GFP–co-transfected HeLa cells (A, B, C) and mCherry-Rab6/Y42A/BicD2-GFP–co-transfected cells (D, E, F), derived from Video 6, Video 7, Video 8, Video 9, Video 10, and Video 11. Different channels are shown: Rab6 (red channel), BicD2 (green channel), and merged image (yellow). Different time points for a zoomed area highlighted with a yellow square are shown to the right of each field. Blue, purple, and white arrows show migrating vesicles that are positive for BicD2 but negative for Rab6. Yellow arrows show immotile Rab6-positive vesicles that are negative for BicD2. A kymograph (far right) shows the migration of vesicles in the 15-μm segment highlighted with a magenta line. The kymograph scale bars on the x-axis measure 2.5 μm, and the scale bars on the y-axis measure 30 s. **(G)** Quantification from the co-localization analysis shows a significant reduction in the co-localization of Rab6(+)/BicD2(+) signal in vesicles in the Y42A condition compared with the WT. *$P$-value = 0.0286, obtained from the Mann–Whitney test (n = 4 independent experiments).

the Golgi to the ER, the minus end–directed dynein is the dominating motor. For post-Golgi trafficking of secretory vesicles to the exocytosis hotspots, which are located at the plus-ends of microtubules, kinesin-1 is the dominating motor, which undergoes a tug-of-war with BicD2/dynein, fine-tuning plus end–directed motility. Most of the Rab6-positive vesicles are transported in the anterograde direction, for which kinesin-1 is the main responsible motor, and there are two populations of Rab6-positive vesicles with speeds of 1.20 ± 0.26 $\mu$m/s and 1.71 ± 0.49 $\mu$m/s, respectively (Serra-Marques et al, 2020). Only a small number of vesicles are transported towards the retrograde direction, for which dynein is the responsible motor (Grigoriev et al, 2007; Lee et al, 2015; Serra-Marques et al, 2020). It should be pointed out that the low mobility of these mutated Rab6-positive vesicles is surprising, because such a large reduction in motility cannot be explained by disrupting retrograde transport alone, because that would only affect a small number of vesicles. Therefore, these data suggest that anterograde transport is affected as well. The large reduction in motility observed for these Rab6 mutants is also surprising, because there are other anchoring proteins besides BicD2 that can recruit kinesin-1 to secretory vesicles (Grigoriev et al, 2007; Mahajan et al, 2019). It should be noted that we cannot exclude that the Rab6 mutations have other effects in addition to disrupting the interaction with BicD2. These mutations could also affect interactions with other Rab6 effectors, as well as the targeting of Rab6-positive vesicles within different compartments of the Golgi apparatus.

In summary, we propose that the Rab6/BicD2 interaction is necessary not only for activation of dynein but also for activation of kinesin-1 motility in the context of the multi-motor complex that facilitates the transport of Rab6-positive vesicles. In the absence of cargo such as Rab6, BicD2 forms a looped, auto-inhibited conformation, in which the dynein binding site is blocked by the cargo-binding domain of BicD2. Binding of cargo such as Rab6 opens the loop and activates BicD2 for dynein binding (Splinter et al, 2012; Liu et al, 2013; Schlager et al, 2014a; McKenney et al, 2014; Terawaki et al, 2015; Urnavicius et al, 2015; McClintock et al, 2018; Sladewski et al, 2018; Cui et al, 2020). It is conceivable that loop opening of BicD2 is also required for binding of kinesin-1 at the coiled-coil domain 2 of BicD2 (Grigoriev et al, 2007) and for activation of kinesin-1 motility. This is in line with a previous study in SMALED2 patient fibroblasts with a pathogenic BicD2/I189F mutation, which demonstrated with a vesicular stomatitis virus G protein reporter assay that BicD2 is required for the trafficking of constitutive secretory cargoes from the trans-Golgi network to the plasma membrane (Rossor et al, 2020).

In line with these results, we have previously shown for reconstituted Nup358/BicD2/dynein/dynactin motor complexes that the cargo-recognition $\alpha$-helix of Nup358 not only is important for binding to BicD2 but also is an important modulator of dynein motility, as mutations of it reduced speed and run length of the motor complexes in single-molecule processivity assays (Gibson et al, 2022). It is conceivable that structurally distinct BicD2/cargo interactions in Rab6$^{GTP}$/BicD2 and Nup358/BicD2 fine-tune the motility for each motor complex individually. Likely additional factors such as tethering of Rab6$^{GTP}$ to membranes or interaction partners such as kinesin-1, which can also bind to membranes via the Dopey1-Mon2 complex (Grigoriev et al, 2007; Mahajan et al,

2019), will have additional modulatory effects on dynein motility and motor complex formation.

Vesicle-embedded Rab6$^{GTP}$ is part of a motor complex that contains BicD2/dynein and kinesin-1 (Grigoriev et al, 2007; Serra-Marques et al, 2020). Furthermore, kinesin-3 (Kif13B), which is recruited by an unknown adapter, is also important for these vesicles to reach the plus-end of microtubules, to which kinesin-1 binds poorly. This is important because the exocytosis hotspots are located near the dynamic microtubule plus-ends, which are attached to the cell cortex by a complex that contains among other components the Rab6 effector ELKS. ELKS is important to capture vesicles at the exocytosis hotspots and promote exocytosis (Grigoriev et al, 2007; Patwardhan et al, 2017; Fourriere et al, 2019). Our structural model suggests that BicD2 competes with ELKS for binding to Rab6. Before ELKS would bind, BicD2 would be released from Rab6-positive vesicles near the plus-ends of microtubules, which would also result in at least partial co-depletion of dynein and kinesin-1 from these vesicles, because they are bound to BicD2. Subsequently, the vesicles will then be recruited to secretion hotspots on the membrane that will be marked by microtubule plus-ends and the associated complex that contains ELKS (Grigoriev et al, 2007; Patwardhan et al, 2017; Fourriere et al, 2019). Our structural model thus suggests additional insights into how Rab6$^{GTP}$ modulates secretion. Overall, all Rab6 effectors studied here, including BicD2, ELKS, the dynactin subunit p150$_{glued}$, Kif20A, GCC185, and R6IP1 (Misery-Lenkei et al, 2007; Burguete et al, 2008; Bergbrede et al, 2009; Recacha et al, 2009; Jin et al, 2023), likely compete for binding to Rab6 and are likely organized as a cascade of effector interactions rather than binding simultaneously to Rab6. However, it should be noted that Rab6 can dimerize, and the possibility exists that distinct interacting partners could bind to distinct Rab6 molecules within the same oligomeric complex.

Our model also provides a basis for regulation of the Rab6/BicD2 interaction by kinases. BicD2 is recruited to Rab6 in the G1 and S phases of the cycle, but in the G2 phase, it localizes to the nuclear envelope and interacts with Nup358 (Splinter et al, 2010). This switch is likely caused by G2 phase–specific kinases, as PLK1 phosphorylates Rab6 on residue S52, lowering the affinity of BicD2 and dynactin to Rab6, and Cdk1 and PLK1 phosphorylate Nup358 in the G2 phase, increasing its affinity to BicD2 (Baffet et al, 2015; Gallisà-Suñé et al, 2023; Jimenez et al, 2023 *Preprint*). In our structural model, Rab6 residue S52 makes a contact with BicD2. Phosphorylated S52 could form intramolecular interactions with nearby Rab6 interface residues R63 and Q65, thereby weakening the interaction with BicD2. Future experiments will establish the mechanism of how BicD2's affinity to distinct cargoes is regulated by phosphorylation.

Several human disease mutations causing spinal muscular atrophy and other neuromuscular diseases are located in the cargo-binding site of BicD2 and affect the affinity to distinct cargoes in a distinct manner. Two mutations were identified that diminish binding to Nup358 but increase the affinity to Nesprin-2 (Yi et al, 2023). Of these two mutations, E774A does diminish binding to Rab6$^{GTP}$, whereas R747C/F743I does not impact Rab6$^{GTP}$ binding (Noell et al, 2019). E774 is an interface residue that interacts with Rab6$^{GTP}$ in our structure, whereas R747 and F743 are located

N-terminally of the Rab6[GTP] binding site. Another disease mutation that causes arthrogryposis multiplex congenita, R690C, selectively increases the affinity to Nup358 fourfold, but Nesprin-2 and Rab6[GTP] binding is not affected (Yi, 2022; Yi et al, 2023). Interestingly, the disease mutant causes defects in Golgi morphology and in neuronal migration; thus, it is conceivable that Nup358 outcompetes Rab6 and Nesprin-2 for binding to BicD2 because of the increased affinity (Yi et al, 2023). Because BicD2 cargoes bind to distinct but overlapping binding sites, the structural characterization of distinct BicD2/cargo complexes is necessary to understand the underlying disease causes of these mutations.

To conclude, here we establish a structural model for recognition of Rab6[GTP] by BicD2. The BicD2 binding site on Rab6 consists of an α-helix, a β-strand, and a coil region and is structurally distinct from the cargo-recognition α-helix that forms the BicD2 binding site in Nup358. However, the binding sites on BicD2 overlap, explaining why Nup358 and Rab6[GTP] compete for binding. The binding site of BicD2 spans the Switch 1 and Switch 2 regions of Rab6 that undergo structural changes in the GTP-bound state. Several hydrophobic interface residues are rearranged in Rab6 upon the transition from the inactive GDP-bound to the active GTP-bound state, explaining why the active state has a higher affinity. Several mutants that disrupt the interaction between Rab6 and BicD2 result in severely impaired motility of Rab6-positive vesicles in cells, suggesting that the interaction between Rab6 and BicD2 is important for activation of the multi-motor complex that includes dynein and kinesin-1. Our results provide new insights into trafficking of secretory and Golgi-derived vesicles for which Rab6 serves as an identity marker and suggest that the Rab6/BicD2 interaction is crucially important for motility of these vesicles, which is important for protein secretion, receptor signaling, and neurotransmission. Our point mutants that target the interaction between Rab6 and BicD2 will enable future studies to establish the role of Rab6 in vesicle transport and secretion. Our results will enable future studies into how these pathways are regulated by phosphorylation through kinases and how these transport pathways are affected by BicD2 human disease mutations that selectively impact binding affinities to distinct cargoes including Rab6.

# Materials and Methods

### GST pull-down assays

All expression constructs were cloned and codon-optimized for expression in *E. coli* as described by the company GenScript, which also performed site-directed mutagenesis (Gibson et al, 2022). The BicD2-CTD expression construct with the N-terminal His$_6$-tag and the thrombin cleavage site was previously described and contains residues 715–804 of human BicD2 cloned into the pet28a vector (Gibson et al, 2022). The expression construct for full-length human Rab6a/Q72L with the N-terminal GST tag that can be cleaved off by PreScission protease in the pGEX6P1 vector was previously described (Gibson et al, 2023). The Q72L mutant was used as it locks Rab6 in the GTP-bound state and renders it GTPase-deficient (Martinez et al, 1994; Matanis et al, 2002). The full-length GTP-bound Rab6a/Q72L is referred to as Rab6[GTP] WT in the study.

Rab6[GTP] and BicD2-CTD constructs were expressed in the *E. coli* BL20(DE3)-RIL strain at 37°C as described.

GST pull-down assays of full-length GST-tagged Rab6[GTP] and BicD2-CTD were performed as described (Gibson et al, 2023). BicD2-CTD WT (residues 715–804) was purified by a single Ni-NTA affinity chromatography step from 1 liter of cell culture. Rab6[GTP] was purified by glutathione Sepharose from 0.5 liter of cell culture, and washed, but not eluted. 1 mM GTP and 2 mM MgCl$_2$ were added, and the columns were incubated for 30 min. Purified BicD2-CTD was added to the columns with bound Rab6[GTP] and incubated for 30 min. The columns were washed and eluted with glutathione elution buffer as described (Cui et al, 2018, 2020; Gibson et al, 2022, 2023). The elution fractions were analyzed by SDS–PAGE, using gels with 16% acrylamide, and stained with Coomassie blue. The gel band intensities of Rab6[GTP] and BicD2-CTD were quantified, and the background intensities were subtracted using ImageJ (Schneider et al, 2012) as described (Yi et al, 2023). The ratio of bound BicD2-CTD/Rab6[GTP] was calculated and normalized to the WT (WT = 1).

### Structure predictions by ColabFold/AlphaFold2

Structure predictions were carried out with the software ColabFold v1.3.0 (Mirdita et al, 2022), which combines the homology search of MMseqs2 with AlphaFold2-multimer (Jumper et al, 2021; Evans et al, 2022 *Preprint*), in the Google Colab AlphaFold2_mmseqs2 Notebook (https://colab.research.google.com/github/sokrypton/ColabFold/blob/main/AlphaFold2.ipynb, accessed on 26 Sep 2022). Amber relaxation was activated; thus, each round of the AlphaFold2 prediction process involved an energy minimization step using the AMBER99SB force field (Hornak et al, 2006), with additional harmonic restraints to maintain the system near the input structure. The restraints are applied independently to heavy atoms, with force constants of 10 kcal/mol Å$^2$ (Jumper et al, 2021).

A 2:2 hetero-tetramer composed of two molecules of human Rab6a/Q72L (residues 13–174 or full-length) and two molecules of human BicD2-CTD (residues 715–804 or full-length) was predicted, using the structure coordinates with the PDB ID 2GIL as a template for Rab6[GTP] (Bergbrede et al, 2005). It was previously established that the Rab6[GTP]/BicD2-CTD complex forms a 2:2 hetero-tetramer (Noell et al, 2018) and that residues 13–174 of Rab6[GTP] have the same affinity towards the BicD2-CTD as the full-length Rab6[GTP] protein, suggesting that they contain the entire BicD2-CTD binding site (Noell et al, 2018).

Other structures were predicted as 2:2 complexes from the sequences, using the same protocol. For the prediction of the full-length Rab6[GTP]/BicD2 complex and the prediction of the Rab6[GTP]/p150$_{glued}$ complex, ColabFold v1.5.2 was accessed on 6/3/2023 and 6/20/2023, respectively.

Structure figures were created by UCSF Chimera and UCSF ChimeraX (Pettersen et al, 2021). Interface residues were identified in structure coordinates with the PDBePISA server, which identifies atoms that are exposed to the other protein molecule rather than the solvent (Krissinel & Henrick, 2007). Results were visualized in UCSF Chimera (Pettersen et al, 2021). Multiple-sequence alignments were performed with T-Coffee version 11 accessed on 8 Oct 2023 (Di Tommaso et al, 2011). Adobe Photoshop was used for figure preparation.

### CD spectroscopy

CD spectroscopy was performed as described (Gibson et al, 2022). For these experiments, Rab6[GTP] was purified by glutathione Sepharose, as described above, but without the addition of GTP. Rab6[GTP] was eluted from the column by proteolytic cleavage with PreScission protease (Cytiva) to remove the GST tag (i.e., not by glutathione). Purified Rab6[GTP] was incubated for 30 min with 1 mM GTP, and transferred into a buffer of 150 mM NaCl, 2 mM $MgCl_2$, 10 mM Tris, pH 8.0, and 0.2 mM TCEP by three cycles of dilution and concentration. Rab6 was concentrated to 0.3 mg/ml and flash-frozen in liquid nitrogen as described (Gibson et al, 2022). For Rab6[GDP], 1 mM GDP was added instead of GTP, and for nucleotide-free Rab6, no nucleotide was added.

CD data (CD, HT, and absorbance) were recorded with a Jasco J-1100 CD spectrometer from 250 to 190 nm at 10°C. The quartz cuvette had a pathlength of 0.1 cm. The following parameters were used: data pitch: 0.1 nm; D.I.T.: 2 s; bandwidth: 1.00 nm; scanning speed: 50 nm/min; and accumulations: 8. The buffer baseline was subtracted from the CD wavelength scans, and the raw ellipticity Θ (mdeg) was converted to mean residue molar ellipticity (Θ). The protein concentration used for the conversion was determined from the buffer-subtracted absorbance recording of the CD data at 214 nm, using the extinction coefficient 371,769 $M^{-1}$ $cm^{-1}$ (Kuipers & Gruppen, 2007). The data were not smoothened. Three CD wavelength scans from distinct purification batches were recorded for each protein, and representative spectra are shown. The secondary structure content of CD spectra was estimated with the program BeStSel (v1.3.230210, accessed at https://bestsel.elte.hu/index.php on 03/11/2024) (Micsonai et al, 2022).

### Live-cell imaging

HeLa cells were cultured in DMEM supplemented with 10% FBS at 37°C with 5% $CO_2$. Rab6 mutants and the WT were codon-optimized for expression in human cells and cloned into the pmCherry-C1 vector using the EcoRI/BamHI sites (GenScript). The BicD2-GFP plasmid was described in Hu et al (2013). Transient transfections were performed using Effectene (301425; QIAGEN) according to the manufacturer's protocol. For live-cell imaging, 18 h after the transfection, cells were imaged using IX83 Andor Revolution XD Spinning Disk Confocal System with an environmental chamber at 37°C, a 60x oil objective (NA 1.30), and a 2x magnifier coupled with iXon Ultra 888 EMCCD Camera. Images were taken at the rate of one frame per second for 3 min.

ImageJ version 2.14.0 was used (Schindelin et al, 2012; Schneider et al, 2012). Kymographs were generated using the Multi Kymograph 3.0.1 ImageJ (NIH) plug-in. The quantification of the ratio of co-localization was done in the portion of the image considered for the kymographs using the Coloc2 3.0.6 ImageJ (NIH) plug-in.

The number of motile Rab6-positive vesicles was quantified in 1-min videos (1 frame/s), using the Mosaic Particle Tracker 1.6 ImageJ (NIH) plug-in (Sbalzarini & Koumoutsakos, 2005), considering the following parameters: radius: 6; cutoff: 0; percentile: 0.1; displacement: 10; link range: 2; and trajectories: longer than 10 frames. All quantifications are the result of at least three separate experiments. Comparisons between experimental groups were done using a non-parametric test of Kruskal–Wallis for multiple groups and Mann–Whitney for two groups considering significant differences when the P-value was less than 0.05.

For immunofluorescence of fixed cells, cells were fixed with 4% PFA for 20 min, then permeabilized with Triton X-100 for 10 min. A blocking step was done by incubating with donkey normal serum (017-000-001; Jackson ImmunoResearch Laboratories) for 1 h. Then, incubation with the primary antibody (ab167453; Anti-mCherry, Abcam) was done for 2 h, followed by incubation with the secondary antibody (Cy3-AffiniPure Donkey Anti-Rabbit IgG (H+L); 711-165-152; Jackson ImmunoResearch Laboratories) for 1 h. Antibodies were diluted 1:200 in a blocking solution. Three washes of 10 min each were done after each antibody incubation. DAPI (9564; Sigma-Aldrich) was included in the second wash of the secondary antibody in a dilution of 1:10,000. Cells were mounted using Aqua-Poly/Mount (18606; Polysciences) on glass for imaging.

## Data Availability

This study includes no data deposited in external repositories.

## Supplementary Information

## Acknowledgements

We thank Erion Sulaj for the initial characterization of Rab6[GTP] by CD spectroscopy. We thank Julien Robert-Paganin, Rezan Solmaz, and Julie Yi for helpful discussions. SR Solmaz was supported by NIH grant R01 GM144578. This work used the Google Colab AlphaFold2_mmseqs2 Notebook (https://colab.research.google.com/github/sokrypton/ColabFold/blob/main/AlphaFold2.ipynb) and the COSMIC[2] server, which is supported by the National Science Foundation under Award Number 1759826. The CD instrument was supported by NIGMS grants 1R01GM125853-02S1 and 3R35GM130207-01S1.

### Author Contributions

X Zhao: conceptualization, data curation, formal analysis, validation, investigation, visualization, methodology, and writing—original draft, review, and editing.

S Quintremil: conceptualization, data curation, formal analysis, investigation, visualization, methodology, and writing—original draft, review, and editing.

ED Rodriguez Castro: data curation, formal analysis, and writing—review and editing.

H Cui: data curation and formal analysis.

D Moraga: data curation and formal analysis.

T Wang: data curation and formal analysis.

RB Vallee: conceptualization, funding acquisition, validation, and writing—review and editing.

SR Solmaz: conceptualization, resources, data curation, formal analysis, supervision, funding acquisition, validation, investigation,

visualization, methodology, project administration, and writing—original draft, review, and editing.

## Conflict of Interest Statement

The authors declare that they have no conflict of interest.

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
