## [Reviewer comments · Life Science Alliance]

Life Science Alliance

Molecular mechanism for recognition of the cargo adapter Rab6GTP by the dynein adapter BicD2

Xiaoxin Zhao, Sebastian Quintremil, Estrella Rodriguez Castro, Heying Cui, David Moraga, Tingyao Wang, Richard Vallee, and Sozanne Solmaz

DOI: <https://doi.org/10.26508/lsa.202302430>

Corresponding author(s): Sozanne Solmaz, Binghamton University

Review Timeline:

Submission Date:	2023-10-11
Editorial Decision:	2023-11-09
Revision Received:	2024-03-21
Editorial Decision:	2024-04-12
Revision Received:	2024-04-24
Accepted:	2024-04-25

Transaction Report:

November 9, 2023

Re: Life Science Alliance manuscript #LSA-2023-02430-T

Dr. Sozanne R Solmaz
Binghamton University, Department of Chemistry
PO BOX 6000
Binghamton, NY 13902

Dear Dr. Solmaz,

Thank you for submitting your manuscript entitled "Molecular mechanism for recognition of the cargo adapter Rab6GTP by the dynein adapter BicD2" to Life Science Alliance. The manuscript was assessed by expert reviewers, whose comments are appended to this letter. We invite you to submit a revised manuscript addressing the Reviewer comments.

Thank you for this interesting contribution to Life Science Alliance. We are looking forward to receiving your revised manuscript.

Sincerely,

B. MANUSCRIPT ORGANIZATION AND FORMATTING:

Reviewer #1 (Comments to the Authors (Required)):

The manuscript by Zhao et al. investigates the molecular interaction between the Golgi vesicle-associated G protein Rab6 and the dynein activating adaptor BicD2. The authors start from an AlphaFold2-generated prediction of the Rab6[GTP]-BicD2 complex structure and attempt to validate it with biochemical and cellular assays. They also perform comparisons of the features of Rab6 that mediate interactions with BicD2 and other effectors. The paper is well written overall and constitutes a significant amount of experimentation. Whilst generating AlphaFold2 predictions is very straightforward, the validation of the structure is a useful addition to the literature, particularly for specialists working on these proteins. However, there are a number of issues with the manuscript that need addressing before we can recommend publication.

Major points:

1. There is no quantification for Figures 5 and 6. One therefore cannot tell if the reported effects are robust. The authors need to devise and implement suitable metrics. For example, for Figure 5 the authors could consider using the proportion of Rab6-positive vesicles that are motile per cell (with multiple cells analysed per construct) and for Figure 6 one could look at the proportion of Rab6-positive structures that localise with BicD2 per cell (or vice versa; with multiple cells analysed per construct). It would also be helpful to quantify the number of Rab6-positive vesicles for the wild-type and mutant as this could influence the degree of co-localisation in the data presented in Figure 6.
2. Similarly, if the authors want to retain the claim that disrupting the Rab6-BicD2 interaction is associated with impaired minus and plus end directed motion, they need to analyse both retrograde and anterograde movements quantitatively (as best they can given a likely complex arrangement of microtubules) with the wild type and mutant Rab6.
3. Whilst the data in Figures 5 and 6 (assuming they are supported by quantitative analysis) are certainly consistent with the described Rab6-BicD2 interaction being required for Rab6 motility, it seems that the authors cannot rule out the mutated Rab6 residue disrupting the interaction of Rab6 with another effector that causes these effects. We believe the authors should include this caveat in the manuscript. For example, whilst we can see that the mutation does not disrupt association of Rab6 with the Golgi, it could be that it affects targeting to vesicles. This specific possibility should also be mentioned.

Minor points:

1. Some of the terminology in Figure 2 and the associated text (e.g. page 13) is potentially confusing. The terms 'interface residues' and 'not at the interface' seem to be used to denote residues that are and are not required for the interaction. Residues could be at the interface but have a negligible contribution to binding affinity. For example, S52 is defined as 'not at the interface' in Figure 2 whereas in the text it is speculated that it could be involved in a salt bridge interaction when phosphorylated. Similarly, in Figure 2C, D49 is listed as an 'interface residue' but this is potentially misleading as the authors present compelling evidence that the mutation leads to Rab6 misfolding. In our opinion, it would be preferable to refer to residues based on their effect on binding (or better describe the terms they use in the text on page 13).
2. The authors seem to favour a 'cascade' or interactions when considering that Rab6 uses overlapping residues to bind BicD2, dynactin and other effectors. Have the authors considered the alternative possibility that multiple Rab6 molecules are engaged with interactions with the BicD2-dynein-dynactin complex and other factors?
3. Is the AlphaFold model compatible with the single lysine-lysine cross-link seen with the fly Rab6-BicD complex in Liu et al. 2013?
4. Page 4. Change 'MT' to 'microtubule'
5. Page 10. 'Nup358 and BicD2 compete for binding to BicD2'. Presumably this is a typo.
6. Although the writing is generally clear, some parts of the text could be made more concise. E.g., page 10. 'These results are in line with the observation that BicD2 binds to Rab6GTP with 10-fold higher affinity compared to the GDP-bound state (Bergbrede et al, 2009).' This statement is repeated three times (also in pages 7 and 8). The authors should read through the

rest of the text again and see if they can limit redundancy.

7. Page 13. Mention that the Q-to-L mutant protein is used for Rab[GTP]

8. Page 18. Change '...provide new insights how BicD2 selects its cargo for transport' to '...provide new insights into how BicD2 selects its cargo for transport'

9. Page 21. Change 'microtubules plus ends' to 'microtubule plus ends'

10. Page 23. Change 'Our results will enable future studies how these pathways...' to 'Our results will enable future studies into how these pathways...'

11. Human protein names should be written in all caps, and human protein names in all caps but italic.

12. In the text, 'least squares' and 'least-squares' are both used. Choose one for consistency.

13. Page 33. Figure 1 needs a legend title.

Referee cross commenting:

It seems that we are in agreement that the microscopy work needs quantifying robustly and that the description of the interfacial residues and how they informed the experiments needs to be much improved.

Reviewer #2 (Comments to the Authors (Required)):

In this manuscript, the authors have modeled Rab6A/BicD2 complex structure using AlphaFold2 (AF) and validated the binding interface by mutational analysis. Many studies have highlighted the importance of the interaction between Rab6 and BicD2 in transport events at the level of the Golgi complex. This interaction is also important for the development of brain and muscle. Still, the binding sites of BicD2 on Rab6 were not identified. This study is therefore of great potential interest.

The AF model of the complex looks reasonable: the confidence of the prediction is high and the predicted local-distance test score indicates a correct prediction of the inter-subunit and inter-protein organization.

My main concerns relate to the choice of residues analyzed and the circular dichroism (CD) data.

1. The interface between Rab6 and BicD2 was analyzed based on the predicted structure. However, it is not clear how did the authors define the interface residues and why the set of the interface residues in the table Fig. S3D (list of contact residues from the predicted structure) doesn't match the set of the mutated residues analyzed by pull-down assays and circular dichroism (CD) wavelength scans shown in Fig. 2.

For example:

- Y35 is a contact residue from the structure, but it was not analyzed by mutation/pull-down and CD.

- F38 is NOT a contact residue from the structure, but it was analyzed by mutation/pull-down and CD, and also tested in cells. F38 is a guanine moiety binding G1-motif residue (conserved aromatic residue in G-proteins), so the F38A mutation is expected to affect GTP binding. As one can see in the Fig. 3, the residue is far from the bound BicD2 molecule, so it can't be an interface residue. Most probably, this point mutation disrupts BicD2 binding (by pull-down) due to the defective GTP-binding.

- The second mutant tested in cells, Y42A, is also NOT a contact residue from the structure. The residue side chain is forwarded to the bound GTP (Fig. 3) and it contacts the GTP gamma phosphate moiety. The Y42A mutation might make the GTP-Rab6 switch-I less rigid and affect the effector binding not directly.

Consequently, the conclusions from the cell-based assays on the Rab6-F38A and Rab6-Y42A effects (Figs. 5 and 6) could be misleading.

2. In general, a CD spectrum allows to identify secondary structure composition of a protein. A percentage of alpha-, beta- and loop-elements in the structure can be calculated from a CD spectrum curve. If a mutant and the WT protein have the same distribution (percentage) of the alpha-, beta- and loop-elements, they could be considered as being folded in the same way. The authors didn't quantify the secondary structure composition of the WT-Rab6 and mutant proteins from the CD data. It is difficult to conclude on the WT/mutant protein fold similarities based on the apparent CD-curves similarities. And the CD spectra curves do not look very similar for the WT and some of the mutants. WT-Rab6 CD spectra in different plots also look somehow different. Does it represent a structural heterogeneity of the WT Rab6? Or Rab6 instability in the used experimental conditions? It is also surprising to expect a correctly folded Rab6 with mutations affecting residues participating in the nucleotide and Mg²⁺ binding. As mentioned above, F38 is a guanine moiety binding G1-motif, and T45 directly coordinates Mg²⁺.

Overall, I would propose to redefine the list of the interface/contact residues in Rab6 and selecting the residues whose side chains are facing the bound BicD2 in the AF structure. Mutations affecting the nucleotide and Mg²⁺ binding to the GTPase should be avoided. In my opinion, the comparison of the WT-Rab6 and the mutants CD-spectra is also not very conclusive and

requires more work.

Other comment:

3. Live cell imaging (Figs. 5 and 6). These experiments need to be quantified: what is the percentage of mobile/immobile vesicles (Fig. 5), what is the percentage of Rab6 + and BicD2 + vesicles (Fig. 6); what is the average speed of Rab6 Y42A + vesicles compared to WT Rab6 (Fig. 6)?

Response to Reviewer's Comments**Decision letter**

November 9, 2023

Re: Life Science Alliance manuscript #LSA-2023-02430-T

Dr. Sozanne R Solmaz
Binghamton University, Department of Chemistry
PO BOX 6000
Binghamton, NY 13902

Dear Dr. Solmaz,

Thank you for submitting your manuscript entitled "Molecular mechanism for recognition of the cargo adapter Rab6GTP by the dynein adapter BicD2" to Life Science Alliance. The manuscript was assessed by expert reviewers, whose comments are appended to this letter. We invite you to submit a revised manuscript addressing the Reviewer comments.

Thank you for this interesting contribution to Life Science Alliance. We are looking forward to receiving your revised manuscript.

Sincerely,

- A letter addressing the reviewers' comments point by point.
- An editable version of the final text (.DOC or .DOCX) is needed for copyediting (no PDFs).
- High-resolution figure, supplementary figure and video files uploaded as individual files: See our detailed guidelines for preparing your production-ready images, <https://www.life-science-alliance.org/authors>
- Summary blurb (enter in submission system): A short text summarizing in a single sentence the study (max. 200 characters including spaces). This text is used in conjunction with the titles of papers, hence should be informative and complementary to the title and running title. It should describe the context and significance of the findings for a general readership; it should be written in the present tense and refer to the work in the third person. Author names should not be mentioned.
- By submitting a revision, you attest that you are aware of our payment policies found here: <https://www.life-science-alliance.org/copyright-license-fee>

B. MANUSCRIPT ORGANIZATION AND FORMATTING:

RESPONSE TO REVIEWER 1:

Reviewer #1 (Comments to the Authors (Required)):

The manuscript by Zhao et al. investigates the molecular interaction between the Golgi vesicle-associated G protein Rab6 and the dynein activating adaptor BicD2. The authors start from an AlphaFold2-generated prediction of the Rab6[GTP]-BicD2 complex structure and attempt to validate it with biochemical and cellular assays. They also perform comparisons of the features of Rab6 that mediate interactions with BicD2 and other effectors. The paper is well written overall and constitutes a significant amount of experimentation. Whilst generating AlphaFold2 predictions is very straightforward, the validation of the structure is a useful addition to the

literature, particularly for specialists working on these proteins. However, there are a number of issues with the manuscript that need addressing before we can recommend publication.

Major points:

COMMENT: 1. There is no quantification for Figures 5 and 6. One therefore cannot tell if the reported effects are robust. The authors need to devise and implement suitable metrics. For example, for Figure 5 the authors could consider using the proportion of Rab6-positive vesicles that are motile per cell (with multiple cells analysed per construct) and for Figure 6 one could look at the proportion of Rab6-positive structures that localise with BicD2 per cell (or vice versa; with multiple cells analysed per construct). It would also be helpful to quantify the number of Rab6-positive vesicles for the wild-type and mutant as this could influence the degree of co-localisation in the data presented in Figure 6.

RESPONSE: We added the quantification results as requested. For Fig. 5, we added the quantification of motile Rab6-positive vesicles for WT Rab6, as well as for the F38A and the Y42A mutants (Fig. 5D and Fig. S11). The quantification confirms that the motility of Rab6-positive vesicles is significantly reduced in the F38A and Y42A mutants compared to the WT. For Fig. 6, we quantified the colocalization of Rab6 and BicD2 (Fig. 6G), and the results confirm that it is significantly decreased in the Y42A mutant compared to the WT.

New panel Fig. 5D

(D) Quantification of the number of Rab6-positive vesicle trajectories identified during live imaging in cells expressing fluorescent fusion proteins of Rab6-WT, Rab6/F38A and Rab6/Y42A. Both mutants show a significant decrease of moving Rab6-positive vesicle trajectories compared to the WT. *P values of 0.0090 for WT vs F38A, and 0.0113 for WT vs Y42A were obtained with the Kruskal-Wallis test (n=3) (see also Fig S11).

New Fig S11

Figure S11. Quantification of moving Rab6-positive vesicles by particle tracking. (A-C) Moving Rab6-positives vesicles are marked by distinctly colored circles. (A) WT; (B) F38A mutant; (C) I42A mutant. (D) A zoom of the area highlighted with the yellow rectangle for the WT condition is shown to illustrate the used particle tracking method. The micrographs show 4 time points from a 1-min video (every 20 seconds), in which the identified trajectories are highlighted by distinctly colored circles (see Fig 6).

Updated Results for Fig. 5D:

“To quantify the effect of these mutants on the motility of the Rab6-positive vesicles we determined the number of trajectories of Rab6-positive vesicles for the WT, the F38A mutant and the Y42A mutant by using automated particle tracking that identifies the trajectories of moving particles (Fig 5D, Fig S11). A recent paper concluded that the results obtained from this automated analysis were very similar to those determined manually from the analysis of kymographs drawn along Rab6-positive tracks (Serra-Marques *et al*, 2020; Schlager *et al*, 2014b; Grigoriev *et al*, 2007). The results of our analysis showed that the number of motile vesicle trajectories is significantly reduced in both the F38A mutant and the Y42A mutant compared to the WT.”

New Fig. 6G

(G) The quantification from the colocalization analysis shows a significant reduction in the colocalization of Rab6(+)/BicD2(+) signal in vesicles in the Y42A condition compared to the WT. * P value = 0.0286, obtained from the Mann-Whitney test (n=4).

Updated results for Fig. 6G:

“We also quantified the ratio of colocalization of Rab6 and BicD2, which is significantly decreased in the Y42A mutant compared to the WT (Fig. 6G).”

Experimental details are described in the **updated Methods** section.

Additional comments:

To address this comment, we quantified the number of motile Rab6-positive vesicles in the cell. We can, however, unfortunately not quantify the number of total Rab6 positive vesicles in cells, since immotile vesicles are difficult to distinguish from other objects that could contain fluorescent Rab6, such as precipitated proteins. Also, the Golgi region has a very dense Rab6 fluorescence that cannot be included in the quantification (see e.g. Serra-Marquez *et al*, 2020).

COMMENT: 2. Similarly, if the authors want to retain the claim that disrupting the Rab6-BicD2 interaction is associated with impaired minus and plus end directed motion, they need to analyse both retrograde and anterograde movements quantitatively (as best they can given a likely complex arrangement of microtubules) with the wild type and mutant Rab6.

RESPONSE: We removed the statement that disrupting the Rab6/BicD2 interaction by point mutations “affects both plus-end and minus-end directed motility on microtubules” from the result section as requested. We also removed the statement regarding the effects on plus and minus end directed motility from the abstract and the discussion session.

COMMENT: 3. Whilst the data in Figures 5 and 6 (assuming they are supported by quantitative analysis) are certainly consistent with the described Rab6-BicD2 interaction being required for Rab6 motility, it seems that the authors cannot rule out the mutated Rab6 residue disrupting the interaction of Rab6 with another effector that causes these effects. We believe the authors should include this caveat in the manuscript. For example, whilst we can see that the mutation does not disrupt association of Rab6 with the Golgi, it could be that it affects targeting to vesicles. This specific possibility should also be mentioned.

RESPONSE: We added the following caveat to the discussion:
“It should be noted that we cannot exclude that the Rab6 mutations have other effects in addition to disrupting the interaction with BicD2. These mutations could also affect interactions with other Rab6 effectors as well as the targeting of Rab6-positive vesicles within different compartments of the Golgi apparatus.”

Minor points:

COMMENT: 1. Some of the terminology in Figure 2 and the associated text (e.g. page 13) is potentially confusing. The terms 'interface residues' and 'not at the interface' seem to be used to denote residues that are and are not required for the interaction. Residues could be at the interface but have a negligible contribution to binding affinity. For example, S52 is defined as 'not at the interface' in Figure 2 whereas in the text it is speculated that it could be involved in a salt bridge interaction when phosphorylated. Similarly, in Figure 2C, D49 is listed as an 'interface residue' but this is potentially misleading as the authors present compelling evidence that the mutation leads to Rab6 misfolding. In our opinion, it would be preferable to refer to residues based on their effect on binding (or better describe the terms they use in the text on page 13).

RESPONSE: We updated the labels in Fig. 2 and the associated text to replace “interface residues” with “Pull-down assays: **Binding reduced**; **Binding not altered**;”. Similar edits were made throughout the manuscript to replace the term “interface residues” with “residues important for the interaction” or “residues important for binding”.

COMMENT: 2. The authors seem to favour a 'cascade' or interactions when considering that Rab6 uses overlapping residues to bind BicD2, dynactin and other effectors. Have the authors considered the alternative possibility that multiple Rab6 molecules are engaged with interactions with the BicD2-dynein-dynactin complex and other factors?

RESPONSE: We updated the discussion section to include this caveat:
“However, it should be noted that Rab6 can dimerize, and the possibility exists that distinct interacting partners could bind to distinct Rab6 molecules within the same oligomeric complex.”

COMMENT: 3. Is the AlphaFold model compatible with the single lysine-lysine cross-link seen with the fly Rab6-BicD complex in Liu et al. 2013?

RESPONSE: Yes. The cross-link was identified between K12 of Rab6 and K730 of *Drosophila* BicD, which corresponds to K789 of *Hs* BicD2. BicD2 residue K789 forms a contact with Rab6 in our AlphaFold2 model and has been established to be important for the interaction (Liu et al., 2013, Terawaki et al., 2015).

The first 13 residues of Rab6 are predicted to be intrinsically disordered and are omitted from the X-ray structure and the AlphaFold2 model presented in Fig. 1. However, we have also included an AlphaFold2 prediction of full-length Rab6 and full-length BicD2 (Fig. S2), which includes residue 12 that forms the cross-link. Based on this prediction, the first 13 residues of Rab6 are intrinsically

disordered. While Alphafold2 can predict disorder with high accuracy, it cannot predict the structure of such intrinsically disordered regions with confidence and typically models them as extended, which does not really reflect the actual structure that would be expected to form a random coil. It should be noted that the disordered region that contains K12 points towards the C-terminal portion of BicD2, therefore the presented Alphafold2 model is consistent with the ability of residue K12 of Rab6 to form a cross-link with K789 of BicD2.

COMMENT: 4. Page 4. Change 'MT' to 'microtubule'

RESPONSE: The text was updated.

COMMENT: 5. Page 10. 'Nup358 and BicD2 compete for binding to BicD2'. Presumably this is a typo.

RESPONSE: Thank you, the text was updated.

COMMENT: 6. Although the writing is generally clear, some parts of the text could be made more concise. E.g., page 10. 'These results are in line with the observation that BicD2 binds to Rab6GTP with 10-fold higher affinity compared to the GDP-bound state (Bergbrede et al, 2009).' This statement is repeated three times (also in pages 7 and 8). The authors should read through the rest of the text again and see if they can limit redundancy.

RESPONSE: We edited the statements on page 8 and 10 and the manuscript was checked for redundancy.

COMMENT: 7. Page 13. Mention that the Q-to-L mutant protein is used for Rab[GTP]

RESPONSE: We added the following sentence: "For these experiments, the Q72L mutant of Rab6 was used as WT, which is locked in the GTP-bound state that has a higher affinity to BicD2-CTD."

COMMENT: 8. Page 18. Change '...provide new insights how BicD2 selects its cargo for transport' to '...provide new insights into how BicD2 selects its cargo for transport'

RESPONSE: The text was updated.

COMMENT: 9. Page 21. Change 'microtubules plus ends' to 'microtubule plus ends'

RESPONSE: The text was updated.

COMMENT: 10. Page 23. Change 'Our results will enable future studies how these pathways...' to 'Our results will enable future studies into how these pathways...'

RESPONSE: The text was updated.

COMMENT: 11. Human protein names should be written in all caps, and human protein names in all caps but italic.

RESPONSE: We assume that reviewer 1 means that human gene names (rather than protein names) should be written in all caps as well as in italics? We updated the human gene names in the manuscript to this nomenclature as requested by the reviewer. However, while we are familiar with the all-caps nomenclature for human proteins, and while we have used it for a recent publication that characterized a human disease mutation in BICD2, such all-caps nomenclature is most commonly used for human genetics studies. It is not very common in the structural biology field to write all human protein names in all caps. We worry that because of the very large number of protein names, the manuscript would become difficult to read for a structural biology audience if we changed the nomenclature of all human proteins to all-caps. For example, Rab6^{GTP}/BicD2/dynein/dynactin would become RAB6GTP/BICD2/DYNEIN/DYNACTIN, which is difficult to read. If the reviewer insists on the change, we will reconsider.

COMMENT: 12. In the text, 'least squares' and 'least-squares' are both used. Choose one for consistency.

RESPONSE: The text was updated to “least-squares”.

COMMENT: 13. Page 33. Figure 1 needs a legend title.

RESPONSE: Fig. 1 legend title: “A structural model of the Rab6^{GTP}/BicD2-CTD complex was obtained from AlphaFold2.”

Referee cross commenting:

It seems that we are in agreement that the microscopy work needs quantifying robustly and that the description of the interfacial residues and how they informed the experiments needs to be much improved.

RESPONSE: See response to Reviewer 1 and Reviewer 2.

RESPONSE TO REVIEWER 2:

Reviewer #2 (Comments to the Authors (Required)):

In this manuscript, the authors have modeled Rab6A/BicD2 complex structure using AlphaFold2 (AF) and validated the binding interface by mutational analysis. Many studies have highlighted the importance of the interaction between Rab6 and BicD2 in transport events at the level of the Golgi complex. This interaction is also important for the development of brain and muscle. Still, the binding sites of BicD2 on Rab6 were not identified. This study is therefore of great potential interest.

The AF model of the complex looks reasonable: the confidence of the prediction is high and the

predicted local-distance test score indicates a correct prediction of the inter-subunit and inter-protein organization.

My main concerns relate to the choice of residues analyzed and the circular dichroism (CD) data.

COMMENT:

1. The interface between Rab6 and BicD2 was analyzed based on the predicted structure. However, it is not clear how did the authors define the interface residues and why the set of the interface residues in the table Fig. S3D (list of contact residues from the predicted structure) doesn't match the set of the mutated residues analyzed by pull-down assays and circular dichroism (CD) wavelength scans shown in Fig. 2. For example:
2. - Y35 is a contact residue from the structure, but it was not analyzed by mutation/pull-down and CD.
- F38 is NOT a contact residue from the structure, but it was analyzed by mutation/pull-down and CD, and also tested in cells. F38 is a guanine moiety binding G1-motif residue (conserved aromatic residue in G-proteins), so the F38A mutation is expected to affect GTP binding. As one can see in the Fig. 3, the residue is far from the bound BicD2 molecule, so it can't be an interface residue. Most probably, this point mutation disrupts BicD2 binding (by pull-down) due to the defective GTP-binding.
- The second mutant tested in cells, Y42A, is also NOT a contact residue from the structure. The residue side chain is forwarded to the bound GTP (Fig. 3) and it contacts the GTP gamma phosphate moiety. The Y42A mutation might make the GTP-Rab6 switch-1 less rigid and affect the effector binding not directly. Consequently, the conclusions from the cell-based assays on the Rab6-F38A and Rab6-Y42A effects (Figs. 5 and 6) could be misleading.

RESPONSE: To address this comment, we performed additional experiments and updated Fig. 2, Fig. S7, Fig. S8, Table S2, Table S3, Table S4, Fig. 5 and Fig. S10.

1. As suggested by the reviewer, we updated the labels in Fig. 2 and the associated text to replace "interface residues" with "Pull-down assays: **Binding reduced**; **Binding not altered**;" . Similar edits were made throughout the manuscript to replace the term "interface residues" with "residues important for the interaction" or "residues important for binding".
2. To address the comment of why Y35A was not characterized, we characterized the mutants of all remaining residues that were identified as contact residues from the AlphaFold2 model by pull-down assays and CD spectroscopy: Y35A, T54A, and L78A. All three mutants reduce binding to BicD2 and are correctly folded according to CD (note that Q72 cannot be mutated as the Q72L mutant is used for these experiments). These results are shown in the updated Fig 2, Fig S7, Fig S8, Table S3 and Table S4.

New data for Fig. S7:

3. To answer the question why in addition to the contact residues from the AlphaFold2 model other residues were mutated as well: Most residues from the Switch 1 and Switch 2 regions of Rab6 were mutated as well, which included F38A and Y42A. The rationale for these additional mutants was that these regions are prime candidates for the binding site because they undergo structural changes when Rab6 transitions to the high affinity GTP-bound state. Furthermore, while the AlphaFold2 model is expected to describe the contact residues of the Rab6/BicD2 complex with a high degree of confidence, I would not expect it to be accurate enough to predict correct residue pairings of salt bridges or hydrogen bonds, or exact orientations of protein side chains. For example, if the error between the true structure and the AlphaFold2 model was 0.5 Angstrom, and Rab6 was shifted by 0.5 Angstrom relative to BicD2, then the contact residues would still be highly accurate, but some of the Rab6 residues may be paired in salt bridges or hydrogen bonds with different residues from BicD2. For that reason, we have included the characterization of additional mutants of the Switch 1 and Switch 2 contact region, as this combined dataset gives a more accurate representation of the true structure. Any residue that is important for the interaction of Rab6 and BicD2 provides valuable information that we would like to include.
4. To answer the question why F38A and Y42A were chosen for the cell biology experiments: While these residues are not AlphaFold2 contact residues, they are in the Switch 1 and Switch 2 region of Rab6 and our binding assays clearly show that these mutants disrupt the interaction of Rab6 with BicD2, while they do not interfere with protein folding. Therefore, these mutants are expected to disrupt the interaction between Rab6 and BicD2 in cells as well, which is confirmed by our assays (Fig. 6). To address the comment that these mutants may result in defective GTP binding, we have added additional data which suggest that nucleotide binding is not affected by these mutations (Fig. 5, Fig. S10). WT Rab6 and the GTP-locked Q72L mutant localize to Rab6-positive vesicles in cells which are visible as discrete puncta (Fig. S10). In contrast, Rab6/T27N which is locked in the GDP-bound state disperses into the cytosol, resulting in a clear and easily detectable phenotype (Fig. S10). Interestingly, the W67A mutant, which is a contact residue in the AlphaFold2 model, also results in a dispersed localization of Rab6, indicating that this mutation affects binding to GTP even though it is not a contact residue of GTP. However, Rab6/F38A and Rab6/Y42A localize in distinct puncta to the Golgi, very similar as observed in the WT, suggesting that their interaction with GTP is not affected by the mutation.
5. To clearly communicate to the readers which Rab6 residues may engage in GTP or Mg^{2+} binding, we added Table S2 that lists all contact residues of Rab6 with GTP and Mg^{2+} (from PDB ID 2GIL). In the updated Fig. 2, all Rab6 residues that form contacts with GTP and Mg^{2+} are now clearly denoted with asterisks and plus signs, which is emphasized in the figure legend. The number of these contact residues is very large and several of these GTP or Mg^{2+} contact residues have no effect on the interaction with BicD2 when mutated to alanine. Therefore, we would like to retain them in the figure, but they are now clearly highlighted. We have also updated the result section to include this caveat:

Updated results:

“Fig. 2C shows the sequence of Rab6, in which the mutated residues that showed reduced binding in the pulldown assays are colored red. In addition, Rab6 residues that form contacts with both GTP and Mg^{2+} are indicated by an asterisk and Rab6 residues that form contacts with GTP are indicated by a plus sign (see Table S2). It should be noted that mutations of GTP or Mg^{2+} contacting residues may possibly alter the interaction of Rab6 with GTP, which would in turn impact binding to BicD2.”

Updated Fig. 2:

Figure 2. The structural model of the Rab6^{GTP}/BicD2-CTD complex is validated by mutagenesis. (A) Least-squares superimposition of the structures of GTP (purple and black)- and GDP-bound (cyan) Rab6b in cartoon representation (Garcia-Saez *et al*, 2006). Switch 1 and Switch 2, two regions that undergo structural changes in the GTP-bound state are colored black in Rab6b^{GTP} (Garcia-Saez *et al*, 2006). **(B,C)** All BicD2-contacting Rab6^{GTP} residues from the AlphaFold2 model and additional residues from the Switch 1 and Switch 2 region were mutated to alanine and binding to the BicD2-CTD was assessed by pulldown assays. The elution fractions were analyzed on SDS-PAGE and the intensities of the gel bands were quantified with ImageJ (Schneider *et al*, 2012). **(B)** Representative SDS-PAGE of elution fractions of GST-pulldowns. Left panel: WT (wild-type), R74A and F75A mutant. Right panel: WT2 and Y42A mutant. Molar masses of standards are indicated on the left. A full representative dataset is shown in Fig S7. Three datasets were collected. An asterisk indicates the position of GST. **(C)** The sequence of Rab6 is shown, residues for which the mutations reduced binding to BicD2 are colored red, residues for which the mutations did not diminish binding are colored blue, and residues that were not assessed are colored white. Rab6 residues that form contacts with both GTP and Mg²⁺ are indicated by an asterisk and Rab6 residues that form contacts with GTP are indicated by a plus sign (see Table S2). Middle panel: Bar graph showing the ratio of bound BicD2/Rab6^{GTP} from pulldown assays normalized respective to the WT (WT=1; reduced binding= red, normal binding = blue). Ratios were averaged from three experiments, and the error bars show the standard deviation. **(D)** The Rab6 mutants that resulted in reduced binding to BicD2 (colored red in C) were characterized by circular dichroism (CD) spectroscopy to assess their secondary structure content. Residues colored white and blue in C were not assessed. Representative CD wavelength scans are shown for the WT (blue) and three mutants (red; K53A, I79A and Y42A). CD wavelength scans for all

mutants are shown in Fig S8. Note that the CD spectra of all 19 mutants that reduced binding were similar to the WT, suggesting that they do not misfold. All CD experiments were performed three times with independently purified samples. See also Table S3 and S4.

New panel Fig. 5E:

(E) Micrograph of immunostained fixed HeLa cells that transiently express a red fluorescent mCherry fusion protein of Rab6/W67A. The Rab6/W67A mutant disperses into the cell, similarly, as observed for the GDP-locked Rab6 mutant (see Fig S10).

New Fig. S10:

Figure S10. Rab6^{GTP} localizes to vesicles of the Golgi apparatus, whereas Rab6^{GDP} disperses into the cell. Micrographs of immunostained fixed HeLa cells that transiently express red fluorescent mCherry fusion proteins of: (A) WT Rab6; (B) the dominant negative (DN) Rab6/T27N mutant which is locked in the GDP-bound state; (C) the constitutively active (CA) Rab6/Q72L mutant, which is locked in the GTP-bound state. Both Rab6 WT and the GTP-locked Rab6 CA localize to vesicles of the Golgi, but the GDP-locked Rab6 DN disperses into the cell. DAPI stain (blue) indicates the location of the nucleus.

Updated results:

“While the number of the motile Rab6-positive vesicles is greatly reduced for the mutants, the Rab6-related fluorescence signal in the Golgi area remains strong, suggesting that these mutations do not affect localization of Rab6 to the Golgi membranes. Since membrane-associated Rab6 is mainly in the GTP-bound state, the strong localization of the mutants and the WT to Golgi-derived Rab6-positive vesicles suggests that the F38A and Y42A mutations do not affect the equilibrium between GDP- and GTP-bound states of Rab6. To compare, Fig S10 shows micrographs of cells expressing fluorescent fusion proteins of WT Rab6 as well as the Rab6/Q72L mutant which is locked in the GTP-bound state, and for both conditions Rab6 localizes robustly to Golgi-derived vesicles, somewhat similar as observed for the F38A and Y42A mutants. To compare, the Rab6/T27N mutant, which is locked in the GDP-bound state disperses to the cytosol, resulting in an easily detectable phenotype. It should be noted that one of the mutants we tested, Rab6/W67A also disperses to the cytosol, suggesting that this mutation could affect the nucleotide-bound state of Rab6 and inactivate it (Fig. 5E). We conclude from these data that the Y42A and F38A mutations do not significantly diminish binding of GTP to Rab6.”

COMMENT: 2. In general, a CD spectrum allows to identify secondary structure composition of a protein. A percentage of alpha-, beta- and loop-elements in the structure can be calculated from a CD spectrum curve. If a mutant and the WT protein have the same distribution (percentage) of the alpha-, beta- and loop-elements, they could be considered as being folded in the same way. The authors didn't quantify the secondary structure composition of the WT-Rab6 and mutant

proteins from the CD data. It is difficult to conclude on the WT/mutant protein fold similarities based on the apparent CD-curves similarities. And the CD spectra curves do not look very similar for the WT and some of the mutants. WT-Rab6 CD spectra in different plots also look somehow different. Does it represent a structural heterogeneity of the WT Rab6? Or Rab6 instability in the used experimental conditions?

It is also surprising to expect a correctly folded Rab6 with mutations affecting residues participating in the nucleotide and Mg²⁺ binding. As mentioned above, F38 is a guanine moiety binding G1-motif, and T45 directly coordinates Mg²⁺.

RESPONSE: For the revision, we optimized our sample preparation, data collection and data processing protocol for CD spectroscopy and acquired a new dataset of CD spectra for all 19 mutants (including the three additional mutants of contact residues from AlphaFold2 that had not been characterized). The optimized protocol resulted in higher quality CD spectra. Key changes of the protocol were the decrease of the protein concentration from 0.65 mg/ml to 0.3 mg/ml, the buffer exchange in the concentration filter instead of dialysis, and the change of data collection parameters (increase of D.I.T, increase of the number of accumulations and decrease of the data pitch to 0.1 nm). Due to the improved data quality, it was unnecessary to smoothen the data. One of the key sources of the error for CD spectroscopy is the concentration determination, which is used for the conversion to molar ellipticity. Therefore, in the updated protocol, we determine the protein concentration from the buffer-subtracted absorbance recording of the CD data at 214 nm, which decreased the overall error. Details can be found in the updated method section for CD spectroscopy. After this optimization process, the CD spectra of all mutants were very similar compared to the WT, including the mutants F38 and T45, which the reviewer mentioned in his/her comments (see updated Fig. S8). It should be noted that the experimental error of the molar ellipticity from CD spectroscopy is 3 - 5% (Cui *et al.*, 2020), therefore the observed differences in the CD spectra are not statistically significant. We also included an overlay of CD spectra of nucleotide-free Rab6, Rab6^{GTP} and Rab6^{GDP}, which suggests that the nucleotide state only results in very minor changes of the CD spectra, which is to be expected based on the published X-ray structures. We also estimated the secondary structure content of all CD spectra with the software BeStSel (Table S3). Finally, we confirmed that the Rab6 samples are structurally stable over time. The spectra of freshly purified Rab6 are comparable to a sample stored for several days at 4°C.

Updated result section:

“In total, the secondary structure content of all nineteen mutants that reduced binding to BicD2 (red in Fig 2C) was assessed by CD spectroscopy. The CD spectra of all nineteen mutants resembled the WT spectra, suggesting that all mutants were correctly folded and did not have significant structural changes compared to the WT (Fig 2D, Fig S8). The secondary structure content of all CD spectra was estimated with the program BeStSel and is summarized in Table S3. It should be noted that some minor changes were observed in some of the CD spectra of the mutants, as well as the derived secondary structure content (Fig S8, Table S3). These differences are not significant, since the experimental error of the molar ellipticity is 3.5 - 5%. We recently determined this error by calculating the standard deviation of the molar ellipticity at 208 and 222 nm from 10 experiments for which the samples were independently prepared (including the determination of the protein concentration, which is the main source of the experimental error) (Cui *et al.*, 2020).”

Updated Fig. S8

Figure S8. Representative CD wavelength scans of purified Rab6^{GTP} mutants. The molar ellipticity is plotted versus the wavelength. Mutant CD wavelength scans (red) are overlaid with CD wavelength scans of the WT protein (blue). Note that the spectra of all mutants that were assessed here resemble the WT, indicating that the mutations do not result in protein misfolding. The final graph shows an overlay of the spectra of nucleotide-free Rab6, Rab6^{GTP} ("WT") and Rab6^{GDP}. See Fig 2. Note that the representative data from Fig 2 is reprinted here. Secondary structure analysis of the spectra is shown in Table S3.

Updated Table S3

See next page

Table S3 Secondary structure estimation from the CD spectra with BestSel (Micsonai *et al*, 2022).

The X-ray structure of human Rab6^{GTP}/Q72L (residues 13-174, PDB ID 2GIL) consists of: 35.1% α -helix; 18.8% β -sheet. These percentages were calculated for the full-length protein with 208 residues (assuming disordered C- and N-termini), since the full-length protein was analyzed by CD spectroscopy.

Rab6^{GTP} mutant	α-helix (%)	Anti-parallel β-sheet (%)	Parallel β-sheet (%)	Turn (%)	Other (%)
WT	21.8	14.7	4.3	15.7	43.4
F75A	20.5	18.2	5.1	15.3	40.9
Y82A	20.3	10.6	3.4	13.7	52.1
I79A	20.4	13.5	5.0	13.1	48.0
P80A	21.2	10.1	4.4	13.6	50.7
I83A	20.2	9.8	5.8	13.8	50.4
Y42A	22.0	12.9	6.8	13.9	44.4
I46A	23.0	11.6	4.2	15.0	46.2
D49A	20.9	11.9	5.7	14.1	47.5
F50A	22.4	12.5	2.1	16.1	46.9
F38A	22.2	11.0	6.3	14.5	46.0
T45A	22.0	10.1	8.4	14.0	45.5
K53A	21.7	11.3	4.6	15.0	47.4
R63A	23.8	10.3	4.7	13.9	47.2
Q65A	22.5	10.3	5.3	13.9	47.9
W67A	22.4	14.6	4.0	16.0	43.0
T69A	21.6	13.8	6.4	15.4	42.8
Y35A	21.4	12.9	5.6	13.9	46.3
T54A	23.8	11.2	4.5	14.7	45.8
L78A	21.5	16.0	3.8	15.7	42.9
Rab6GDP	23.5	10.8	3.2	13.9	48.6
Rab6 nucleotide free	25.1	11.9	4.3	15.6	43.1

COMMENT: Overall, I would propose to redefine the list of the interface/contact residues in Rab6 and selecting the residues whose side chains are facing the bound BicD2 in the AF

structure. Mutations affecting the nucleotide and Mg²⁺ binding to the GTPase should be avoided. In my opinion, the comparison of the WT-Rab6 and the mutants CD-spectra is also not very conclusive and requires more work.

RESPONSE: See responses to the first two comments.

Other comment:

COMMENT: 3. Live cell imaging (Figs. 5 and 6). These experiments need to be quantified: what is the percentage of mobile/immobile vesicles (Fig. 5), what is the percentage of Rab6 + and BicD2 + vesicles (Fig. 6); what is the average speed of Rab6 Y42A + vesicles compared to WT Rab6 (Fig. 6)?

RESPONSE: We added the quantification results as requested. For Fig. 5, we added the quantification of motile Rab6-positive vesicles for WT Rab6, as well as the F38A and the Y42A mutants (Fig. 5D and Fig. S11). The quantification confirms that the motility of Rab6-positive vesicles is significantly reduced in the F38A and Y42A mutants compared to the WT. For Fig. 6, we quantified the colocalization of Rab6 and BicD2 (Fig. 6G), and the results confirm that the colocalization is significantly decreased in the Y42A mutant compared to the WT.

We can, however, unfortunately not quantify the number of immotile Rab6-positive vesicles as they are difficult to distinguish from other objects that could contain fluorescent Rab6, such as precipitated proteins. The velocity of WT Rab6-positive vesicles was recently quantified, and there are two populations of Rab6-positive vesicles with speeds of $1.20 \pm 0.26 \mu\text{m}/\text{sec}$ and $1.71 \pm 0.49 \mu\text{m}/\text{s}$ respectively (Serra-Marquez *et al*, 2020). Since two populations with distinct speeds exist, quantifying velocity differences for the mutants compared to the WT is challenging.

Because of this, we chose the number of motile Rab6-positive vesicles in cells for our quantification, as described in more detail below. Our results show that the number of motile Rab6-positive vesicles is significantly reduced in the Y42A and F38A mutants compared to the WT. We also added the published velocity for the WT Rab6-positive vesicles to the discussion section.

Updated discussion

“Most of the Rab6-positive vesicles are transported in the anterograde direction, for which kinesin-1 is the main responsible motor, and there are two populations of Rab6-positive vesicles with speeds of $1.20 \pm 0.26 \mu\text{m}/\text{sec}$ and $1.71 \pm 0.49 \mu\text{m}/\text{s}$ respectively (Serra-Marques *et al*, 2020).”

The following description of the newly added quantification results was also included in the response to Reviewer 1 and is reprinted here:

New Fig. 5D

(D) Quantification of the number of Rab6-positive vesicle trajectories identified during live imaging in cells expressing fluorescent fusion proteins of Rab6-WT, Rab6/F38A and Rab6/Y42A. Both mutants show a significant decrease of moving Rab6-positive vesicle trajectories compared to the WT. *P values of 0.0090 for WT vs F38A, and 0.0113 for WT vs Y42A were obtained with the Kruskal-Wallis test (n=3) (see also Fig S11).

New Fig S11

Figure S11. Quantification of moving Rab6-positive vesicles by particle tracking. (A-C) Moving Rab6-positives vesicles are marked by distinctly colored circles. (A) WT; (B) F38A mutant; (C) I42A mutant. (D) A zoom of the area highlighted with the yellow rectangle for the WT condition is shown to illustrate the used particle tracking method. The micrographs show 4 time points from a 1-min video (every 20 seconds), in which the identified trajectories are highlighted by distinctly colored circles (see Fig 6).

Updated Results for Fig. 5D:

“To quantify the effect of these mutants on the motility of the Rab6-positive vesicles we determined the number of trajectories of Rab6-positive vesicles for the WT, the F38A mutant and the Y42A mutant by using automated particle tracking that identifies the trajectories of moving particles (Fig 5D, Fig S11). A recent paper concluded that the results obtained from this automated analysis were very similar to those determined manually from the analysis of kymographs drawn along Rab6-positive tracks (Serra-Marques *et al*, 2020; Schlager *et al*, 2014b; Grigoriev *et al*, 2007). The results of our analysis showed that the number of motile vesicle trajectories is significantly reduced in both the F38A mutant and the Y42A mutant compared to the WT.”

New Fig. 6G

(G) The quantification from the colocalization analysis shows a significant reduction in the colocalization of Rab6(+)/BicD2(+) signal in vesicles in the Y42A condition compared to the WT. * P value = 0.0286, obtained from the Mann-Whitney test (n=4).

Updated results for Fig. 6G:

“We also quantified the ratio of colocalization of Rab6 and BicD2, which is significantly decreased in the Y42A mutant compared to the WT (Fig. 6G).”

Experimental details are described in the **updated Methods** section.

April 12, 2024

RE: Life Science Alliance Manuscript #LSA-2023-02430-TR

Dr. Sozanne R Solmaz
Binghamton University
Department of Chemistry
PO BOX 6000
Binghamton, NY 13902

Dear Dr. Solmaz,

Thank you for submitting your revised manuscript entitled "Molecular mechanism for recognition of the cargo adapter Rab6GTP by the dynein adapter BicD2". We would be happy to publish your paper in Life Science Alliance pending final revisions necessary to meet our formatting guidelines.

- please address the remaining Reviewer comments
- please be sure that the authorship listing and order is correct
- please upload your videos as MP4 or MOV, MPG, and AVI files. Provide one file for each video. Videos may be no larger than 10 MB.
- please add the Twitter handle of your host institute/organization as well as your own or/and one of the authors in our system
- please remove supplementary figures from the manuscript file and leave their legends after the legends for the main figures
- we encourage you to revise the figure legends for Figure S5 such that the figure panels are introduced in an alphabetical order
- please add callouts for Figures 3A-B; 6F; S1A-C; S4A-C; S5A-D; S9A-B; S10A-C and S11A-D to your main manuscript text

FIGURE CHECKS:

- please add a scale bar to Figure S10

A. FINAL FILES:

B. MANUSCRIPT ORGANIZATION AND FORMATTING:

Thank you for your attention to these final processing requirements. Please revise and format the manuscript and upload materials within 15 days.

Sincerely,

Reviewer #1 (Comments to the Authors (Required)):

The authors have satisfactorily addressed all my comments. I now recommend publication pending the following minor points being addressed.

(1) It is not clear in the legends to the new panels 5D and 6G to what the 'n' number refers. Is this number of independent experiments or number of cells, for example. This should be stated in each legend

(2) It is not clear why some text is highlighted in yellow in Table S1.

(3) The abstract states that residues important for the Rab6/BicD2 interaction lead to a loss of co-migration of these proteins in vivo. However, there is still residual co-migration in these experiments. The language should be modified, for example to say that the co-migration was reduced.

Reviewer #2 (Comments to the Authors (Required)):

The authors have responded to many of my previous comments satisfactorily and improved the manuscript. However, they did not address my main concern, i.e. that the mutations tested in cells (F38A and Y42A, Figs 5 and 6) are not good candidates for functional validation of structure-based interaction between RAB6 and BicD2. These residues are not in the RAB6/BicD2 interaction site and are conserved in the whole RAB family. The authors argue that the interaction of RAB6 F38A and RAB6 Y42A with GTP is not affected because they localize to the Golgi region of transfected cells. I am afraid that it is a weak argument that does not rule out the possibility that these mutations affect nucleotide binding/release. A more convincing experiment would be to directly measure biochemically the binding affinities of the mutant proteins for GTP/GDP. They also argue that the W67A mutant (new panel Fig. 5E), which is a contact residue predicted by AlphaFold2, affects binding to GTP though it is not a contact residue of GTP. I do not see the point here. In addition, W67 is a part of the conserved RAB hydrophobic triad (Merithew et al, J Biol Chem 2001). It is a strictly conserved residue which participates in the binding of RAB regulatory proteins and effectors. This mutation may thus affect RAB prenylation and membrane recruitment by many means.

April 12, 2024

RE: Life Science Alliance Manuscript #LSA-2023-02430-TR

Dr. Sozanne R Solmaz
Binghamton University
Department of Chemistry
PO BOX 6000
Binghamton, NY 13902

Dear Dr. Solmaz,

Thank you for submitting your revised manuscript entitled "Molecular mechanism for recognition of the cargo adapter Rab6GTP by the dynein adapter BicD2". We would be happy to publish your paper in Life Science Alliance pending final revisions necessary to meet our formatting guidelines.

- please address the remaining Reviewer comments
- please be sure that the authorship listing and order is correct
- please upload your videos as MP4 or MOV, MPG, and AVI files. Provide one file for each video. Videos may be no larger than 10 MB.
- please add the Twitter handle of your host institute/organization as well as your own or/and one of the authors in our system
- please remove supplementary figures from the manuscript file and leave their legends after the legends for the main figures
- we encourage you to revise the figure legends for Figure S5 such that the figure panels are introduced in an alphabetical order
- please add callouts for Figures 3A-B; 6F; S1A-C; S4A-C; S5A-D; S9A-B; S10A-C and S11A-D to your main manuscript text

Response: All of these edits have been made. A point-by-point response of the reviewer's comments is attached below. The twitter handles are @SolmazSozanne and @binghamtonu. We are providing the supplementary movies as AVI files. These movies are already compressed by a factor of 10 from ~500 MB, and we contacted the editorial office, as we were not able to compress these movies further to 10 MB, without introducing artifacts. The editorial office agreed that they would accept the supplemental movies in the provided file sizes that range from 16 - 50 MB.

FIGURE CHECKS:

-please add a scale bar to Figure S10

Response: We added the scale bar to figure S10.

To upload the final version of your manuscript, please log in to your account: <https://lsa.msubmit.net/cgi-bin/main.plex>

A. FINAL FILES:

B. MANUSCRIPT ORGANIZATION AND FORMATTING:

Thank you for your attention to these final processing requirements. Please revise and format the manuscript and upload materials within 15 days.

Sincerely,

Reviewer #1 (Comments to the Authors (Required)):

The authors have satisfactorily addressed all my comments. I now recommend publication pending the following minor points being addressed.

(1) It is not clear in the legends to the new panels 5D and 6G to what the 'n' number refers. Is this number of independent experiments or number of cells, for example. This should be stated in each legend.

Response: n refers to the number of independent experiments. This information was added to the figure legends of Fig 5 and Fig 6.

(2) It is not clear why some text is highlighted in yellow in Table S1.

Response: To address this comment, we updated the title of Table S1: "Table S1. List of Rab6^{GTP} interface Residues with BicD2-CTD and ELKS (yellow: shared contact residues)."

(3) The abstract states that residues important for the Rab6/BicD2 interaction lead to a loss of co-migration of these proteins in vivo. However, there is still residual co-migration in these experiments. The language should be modified, for example to say that the co-migration was reduced.

Response: We edited the abstract as requested. "Mutations of Rab6^{GTP} that abolish binding to BicD2 also result in reduced co-migration of Rab6^{GTP}/BicD2 in cells, validating our model."

Reviewer #2 (Comments to the Authors (Required)):

The authors have responded to many of my previous comments satisfactorily and improved the manuscript. However, they did not address my main concern, i.e. that the mutations tested in cells (F38A and Y42A, Figs 5 and 6) are not good candidates for functional validation of structure-based interaction between RAB6 and BicD2. These residues are not in the RAB6/BicD2 interaction site and are conserved in the whole RAB family. The authors argue that the interaction of RAB6 F38A and RAB6 Y42A with GTP is not affected because they localize to the Golgi region of transfected cells. I am afraid that it is a weak argument that does not rule out the possibility that these mutations affect nucleotide binding/release. A more convincing experiment would be to directly measure biochemically the binding affinities of the mutant proteins for GTP/GDP. They also argue that the W67A mutant (new panel Fig. 5E), which is a contact residue predicted by AlphaFold2, affects binding to GTP though it is not a contact residue of GTP. I do not see the point here. In addition, W67 is a part of the conserved RAB hydrophobic triad (Merithew et al, J Biol Chem 2001). It is a strictly conserved residue which participates in the binding of RAB regulatory proteins and effectors. This mutation may thus affect RAB prenylation and membrane recruitment by many means.

Response: It should be noted that all residues of Rab6 are highly conserved, not just the ones specifically mentioned here (e.g., W67, see multiple sequence alignment, Fig. S3E). There are no candidate residues for mutations that are not highly conserved.

As previously discussed, the F38A and Y42A mutants were chosen for the cell biology experiments because our binding assays clearly show that these mutants disrupt the interaction of Rab6 with BicD2, while they do not interfere with protein folding. Therefore, these mutants are expected to disrupt the interaction between Rab6 and BicD2 in cells as well, which is confirmed by our assays (Fig. 6). While these residues are not AlphaFold2 contact residues, they are in the Switch 1 and Switch 2 region of Rab6, which undergoes structural changes in the GTP-bound state, and therefore test our overall model that the Switch 1 and Switch 2 regions of Rab6 are important for the interaction with BicD2 and for activation of GTP-bound Rab6 for BicD2 binding. These mutations are therefore good candidates to probe the role of the Rab6/BicD2 interaction in modulation of secretory vesicle motility.

To address the comment that these mutants may result in defective GTP binding: our cell biology data suggest that nucleotide binding and activation is not affected by these mutations (Fig. 5, Fig. S10). WT Rab6 and the GTP-locked Q72L mutant localize to Rab6-positive vesicles in cells which are visible as discrete puncta (Fig. S10). In contrast, Rab6/T27N which is locked in the GDP-bound state disperses into the cytosol, resulting in a clear and easily detectable phenotype (Fig. S10), as GDP-bound Rab6 is recognized by the protein GDI and released from the membrane into the cytosol. However, Rab6/F38A and Rab6/Y42A localize in distinct puncta to the Golgi, very similar as observed for the WT, suggesting that their interaction with GTP, their nucleotide release and their activation is not affected by the mutations, as otherwise they would not associate with the Golgi-membranes. We agree that it would be nice to develop an assay to compare the affinity of GTP to the various Rab6 mutants in future studies.

The W67A mutant, which is a contact residue in the AlphaFold2 model, also results in a dispersed localization of Rab6, indicating that this mutation affects either binding to GTP or activation of Rab6, which would then subsequently impact membrane integration. This is to be expected, as W67 is a member of the conserved Rab invariant hydrophobic triad, which is important for stabilization of the active conformation of Rab6^{GTP}, which has the ability to recruit several Rab6 effectors. The disperse localization of the W67A mutant demonstrates that any mutations affecting activation of Rab6 or its interaction with GTP are expected to result in an easily detectable phenotype with disperse localization, whereas a localization of Rab6 to Golgi-membranes suggests that GTP binding, activation and membrane integration are intact.

We also edited the results section, to clarify the role of the invariant hydrophobic triad of Rab6 in effector recognition:

“It should be noted that three of the Rab6 residues that are both contact residues in the AlphaFold2 model and that are also confirmed to be important for binding to BicD2, constitute the invariant hydrophobic triad, which is conserved in the Rab family of proteins and forms a hydrophobic switch region interface: residues F50, W67 and Y82. This invariant hydrophobic triad has been shown to undergo structural changes upon activation in the GTP bound state that are a determinant for effector recognition, and specifically for BicD2, as we show here (Ostermeier & Brunger, 1999; Merithew *et al*, 2001; Dumas *et al*, 1999).”

“It should be noted that one of the mutants we tested, Rab6/W67A, which is a contact residue in the AlphaFold2 model, also disperses into the cell, indicating that the mutation could either affect binding to GTP, or activation of Rab6, which would subsequently impact membrane integration (Fig. 5E). This is to be expected, as W67 is a member of the invariant hydrophobic triad that is conserved in Rab proteins. This triad is involved in the activation mechanism of Rab6, and important for recognition of effectors such as BicD2 (Ostermeier & Brunger, 1999; Merithew *et al*, 2001; Dumas *et al*, 1999). We conclude that the Y42A and F38A mutations do not significantly diminish binding of GTP or activation of Rab6, as it would result in their dispersal from the Golgi-membranes.”

April 25, 2024

RE: Life Science Alliance Manuscript #LSA-2023-02430-TRR

Dr. Sozanne R Solmaz
Binghamton University
Department of Chemistry
PO BOX 6000
Binghamton, NY 13902

Dear Dr. Solmaz,

Thank you for submitting your Research Article entitled "Molecular mechanism for recognition of the cargo adapter Rab6GTP by the dynein adapter BicD2". It is a pleasure to let you know that your manuscript is now accepted for publication in Life Science Alliance. Congratulations on this interesting work.

DISTRIBUTION OF MATERIALS:

Again, congratulations on a very nice paper. I hope you found the review process to be constructive and are pleased with how the manuscript was handled editorially. We look forward to future exciting submissions from your lab.

Sincerely,
